# Gaussian Partial Information Decomposition: Bias Correction and Application to High-dimensional Data

**Praveen Venkatesh[1,2,*], Corbett Bennett[1], Sam Gale[1], Tamina K. Ramirez[3], Greggory Heller[4], Séverine Durand[1], Shawn Olsen[1], Stefan Mihalas[1,†]**

[1]Allen Institute; [2]University of Washington; [3]Columbia University;
[4]Massachusetts Institute of Technology

[*]praveen.venkatesh@alleninstitute.org; [†]stefanm@alleninstitute.org

## Abstract

Recent advances in neuroscientific experimental techniques have enabled us to simultaneously record the activity of thousands of neurons across multiple brain regions. This has led to a growing need for computational tools capable of analyzing how task-relevant information is represented and communicated between several brain regions. Partial information decompositions (PIDs) have emerged as one such tool, quantifying how much unique, redundant and synergistic information two or more brain regions carry about a task-relevant message. However, computing PIDs is computationally challenging in practice, and statistical issues such as the bias and variance of estimates remain largely unexplored. In this paper, we propose a new method for efficiently computing and estimating a PID definition on multivariate Gaussian distributions. We show empirically that our method satisfies an intuitive additivity property, and recovers the ground truth in a battery of canonical examples, even at high dimensionality. We also propose and evaluate, for the first time, a method to correct the bias in PID estimates at finite sample sizes. Finally, we demonstrate that our Gaussian PID effectively characterizes inter-areal interactions in the mouse brain, revealing higher redundancy between visual areas when a stimulus is behaviorally relevant.

## 1 Introduction

Neuroscientific experiments are increasingly collecting large-scale datasets with simultaneous recordings of multiple brain regions with single-unit resolution [1–3]. These experimental advances call for new computational tools that can allow us to probe how multiple brain regions jointly process relevant information in a behaving animal.

Partial Information Decompositions (PIDs) offer a new method for studying how different brain regions carry task-relevant information: they provide measures to quantify the amount of *unique*, *redundant* and *synergistic* information that one region has with respect to another. The information itself could pertain to task-relevant variables such as stimuli, behavioral responses, or information contained in a third region. For example, we may be interested in how much information about a stimulus is communicated or shared (i.e., redundantly present) between two brain regions over time. Or, we might be interested in the extent to which one region's activity uniquely explains that of another, while excluding information corresponding to spontaneous behaviors.

Ideas such as redundancy and synergy have a long history in neuroscience, having been proposed for understanding noise correlations [4] and to understand differences in encoding complexity between different brain regions [5]. PIDs have also been suggested for quantifying how much sensory information is used to execute behaviors [6] and for tracking stimulus-dependent information flows between brain regions [7, 8]. Outside of neuroscience, PID has been used to understand interactions

between different variables in financial markets [9], to quantify the relevance of different features for the purpose of feature selection in machine learning [10], and to define and quantify bias in the field of fair Machine Learning [11].

An important constraint that has limited the broader adoption of PIDs in neuroscience is the computational difficulty of estimating PIDs for high-dimensional data. Many PID definitions that are operationally well-motivated involve solving an optimization problem over a space of probability distributions: the number of optimization variables can thus be exponential in the number of neurons [12]. This has led to the use of poorly motivated PID definitions that are easy to compute (such as the "MMI-PID" of [13], in works such as [9, 14–16]), or limited analyses to very few dimensions [17]. Furthermore, due to the limited exploration of estimators for PIDs, issues such as the bias and variance of estimates have received no attention so far, to our knowledge.

In this paper, we make the following contributions:

1. We provide a new and efficient method for computing and estimating a well-known PID definition called the ∼-PID or the BROJA-PID [18] on Gaussian distributions (Section 3). By restricting our attention to Gaussian distributions, we are able to significantly reduce the number of optimization variables, so that this is just quadratic in the number of neurons, rather than exponential.
2. We present a set of canonical examples for Gaussian distributions where ground truth is known, and show that our method outperforms others (Section 4).
3. We also raise (for what we believe is the first time) the issue of bias in PID estimates, propose a method for correcting the bias, and empirically evaluate its performance (Section 5).
4. Finally, we show that our Gaussian PID estimator closely agrees with ground truth, even on non-Gaussian distributions, and show an example of its use on real neural data (Section 6).

**Related work**

Our method is based on our earlier work [12], where we also examined PIDs for Gaussian distributions. Our current work differs in a few key aspects: (i) we estimate the PID of a different PID definition, the ∼-PID rather than the $\delta$-PID, because the $\delta$-PID does not satisfy a basic property called additivity [19] (defined in Sec. 2); (ii) our current method provides an exact upper bound to the PID definition being computed, rather than an approximate upper bound; (iii) we now consider the problem of estimation, not just computation, and explore the issue of the bias of PID estimates; and (iv) our current method is much faster, and we demonstrate agreement with ground truth at much higher dimensionality.

Several other studies have also considered methods for efficiently estimating PIDs: Banerjee et al. [20] and Makkeh et al. [21] address computing *discrete* PIDs, but their method does not scale to higher dimensions; Pakman et al. [17] estimate PIDs using copulas, but their method would also potentially be computationally prohibitive at high dimensionalities; Liang et al. [22] use convex optimization to directly estimate the ∼-PID for general high-dimensional distributions, but they do not compare with ground truth at high dimensionality or examine bias in their estimates.

## 2 Background: An Introduction to PIDs and the ∼-PID

In this section, we provide an introduction to the concept of partial information decomposition along with an illustrative example. Let $M$, $X$ and $Y$ be three random variables with joint distribution $P_{MXY}$. A PID decomposes the total mutual information between the *message* $M$ and two *constituent* random variables $X$ and $Y$ into a sum of four non-negative components that satisfy [18, 23]:

$$I\big(M;(X,Y)\big) = UI(M:X \setminus Y) + UI(M:Y \setminus X) + RI(M:X;Y) + SI(M:X;Y) \quad (1)$$

$$I(M;X) = UI(M:X \setminus Y) + RI(M:X;Y) \quad (2)$$

$$I(M;Y) = UI(M:Y \setminus X) + RI(M:X;Y) \quad (3)$$

Here, $I(A;B)$ is the *Shannon mutual information* [24] between the random variables $A$ and $B$, and the four terms in the RHS of (1) are respectively the information about $M$ that is (i) *uniquely* present in $X$ and not in $Y$; (ii) *uniquely* present in $Y$ and not in $X$; (iii) *redundantly* present in both $X$ and $Y$ and can be extracted from either; and (iv) *synergistically* present in $X$ and in $Y$, i.e., information which cannot be extracted from either of them individually, but can be extracted from their interaction. For the sake of brevity, we may also refer to these partial information components as $UI_X$, $UI_Y$, $RI$ and $SI$ respectively. Notwithstanding notation, they should all properly be understood to be functions of the joint distribution $P_{MXY}$.

Now, $UI_X$, $UI_Y$, $RI$ and $SI$ consist of four undefined quantities, subject to the three equations in (1)–(3). In addition, they are typically assumed to be non-negative, $RI$ and $SI$ are each constrained to be symmetric in $X$ and $Y$, and the functional forms of $UI_X$ and $UI_Y$ should be identical when exchanging $X$ for $Y$. Despite the number of constraints, many definitions satisfy all of them, each differing in its motivation and interpretation [18, 23, 25–27] (see [27, 28] for a review), and we need to formally define one of these partial information components to determine the other three.

**Example 1.** Before we jump into a specific definition, we provide an intuition into what these terms mean using a simple example. Suppose $M = [A, B, C]$, $X = [A, B, C \oplus Z]$, and $Y = [B, Z]$, where $A, B, C, Z \sim$ i.i.d. Ber(0.5).[1] Then, $X$ has 1 bit of unique information about $M$, i.e., $A$; $Y$ has no unique information; $X$ and $Y$ both have 1 bit of redundant information, i.e., $B$, since it can be obtained from either $X$ or $Y$; and $X$ and $Y$ have 1 bit of synergistic information, i.e., $C$, which cannot be obtained from either $X$ or $Y$ individually (since $C \oplus Z \perp\!\!\!\perp C$), but can only be recovered when both $X$ and $Y$ are known. For more examples on binary variables, we refer the reader to [18]. □

In this manuscript, we consider a definition that we refer to as the $\sim$-PID[2] [18, 25], which is defined below. We chose to build an estimator for *this* definition for two reasons: (i) it is a *Blackwellian PID* definition, i.e., it has well-defined operational interpretations based on concepts from statistical decision theory (e.g., see [18, 27, 29] for details); and (ii) it satisfies many desirable properties (e.g., see [18, 30]), and in particular, a property that we call *additivity of independent components*.

**Definition 1** ($\sim$-PID [18]). *The unique information about $M$ present in $X$ and not in $Y$ is given by*

$$\widetilde{UI}(M : X \setminus Y) := \min_{Q \in \Delta_P} I_Q(M; X \,|\, Y), \tag{4}$$

*where $\Delta_P := \{Q_{MXY} : Q_{MX} = P_{MX},\ Q_{MY} = P_{MY}\}$ and $I_Q(\,\cdot\,;\cdot\,|\,\cdot\,)$ is the conditional mutual information over the joint distribution $Q_{MXY}$. The remaining $\sim$-PID components, $\widetilde{UI}(M : Y \setminus X)$, $\widetilde{RI}(M : X; Y)$ and $\widetilde{SI}(M : X; Y)$, follow from equations (1)–(3).*

**Property 1** (Additivity of independent components). *Suppose $M = [M_1, M_2]$, $X = [X_1, X_2]$, and $Y = [Y_1, Y_2]$, such that $(M_1, X_1, Y_1) \perp\!\!\!\perp (M_2, X_2, Y_2)$. Then, additivity implies that*

$$UI(M : X \setminus Y) = UI(M_1 : X_1 \setminus Y_1) + UI(M_2 : X_2 \setminus Y_2), \tag{5}$$

*and similarly for the other three partial information components, $UI_Y$, $RI$ and $SI$.*

Property 1 stipulates that we should be able to compute the PIDs of two independent systems *separately*, and then add the components across both systems. In effect, additivity implies that the PID of an isolated system should not depend on the PID of another isolated system, making it an intuitive and highly desirable property (see App. A.1 for concrete examples). Of the many PID definitions examined by Rauh et al. [19], only the $\sim$-PID satisfied additivity (as proved in [18]).

## 3 Computing the $\sim$-PID for Gaussian Distributions

The first contribution of this paper is a method to efficiently compute bounds on the $\sim$-PID for jointly Gaussian random vectors $M$, $X$ and $Y$. To be precise, our method computes an upper bound for $\widetilde{UI}_X$ and $\widetilde{UI}_Y$, and lower bounds for $\widetilde{RI}$ and $\widetilde{SI}$. Similar to our earlier work [12], we present a new PID definition that we call the $\sim_G$-PID, which characterizes an upper bound on the unique information of the $\sim$-PID by restricting the optimization space to jointly Gaussian $Q_{MXY}$:

**Definition 2** ($\sim_G$-PID). *Let $P_{MXY}$ be jointly Gaussian. Then, the unique information about $M$ present in $X$ and not in $Y$ is given by*

$$\widetilde{UI}_G(M : X \setminus Y) := \min_{Q \in \Delta_P} I_Q(M; X \,|\, Y), \tag{6}$$

*where $\Delta_P := \{Q_{MXY} : Q_{MXY}$ jointly Gaussian, $Q_{MX} = P_{MX},\ Q_{MY} = P_{MY}\}$ and $I_Q$ is the conditional mutual information over the joint distribution $Q_{MXY}$.*

If the optimal $Q_{MXY}$ in the unrestricted optimization of Definition 1 happens to be Gaussian for some $P_{MXY}$, then the $\sim_G$-PID would be identical to the $\sim$-PID for that $P_{MXY}$. We conjecture

---

[1] i.i.d. stands for "independent and identically distributed"; $X \perp\!\!\!\perp Y$ means $X$ and $Y$ are independent.

[2] This PID is also sometimes referred to as the BROJA PID (after the authors of [18]), or the minimum-synergy PID in the literature. We prefer to use an author-agnostic nomenclature as introduced in our earlier work [12], because this PID was also introduced contemporaneously by [25].

that this happens whenever $P_{MXY}$ is Gaussian: for example, in a similar optimization problem for computing the information bottleneck [31], the optimal distribution is Gaussian whenever $P$ is Gaussian [32, 33]. We provide empirical evidence in favor of this conjecture through a sequence of examples with Gaussian $P$ in Sec. 4, where we recover the ground truth even when $Q$ is restricted to be Gaussian. However, we leave a theoretical examination of this conjecture for future work.

In practical terms, restricting the search space to Gaussian $Q_{MXY}$ reduces the number of optimization variables from being exponential in the dimensionality to quadratic (see Appendix A.2), allowing us to compute the $\sim_G$-PID for much higher dimensionalities of $M$, $X$ and $Y$. In what follows, we show how the optimization problem for the $\sim_G$-PID can be written out in closed-form and then solved using projected gradient descent.

### 3.1 Notation and Preliminaries

Suppose $M$, $X$ and $Y$ are jointly Gaussian random vectors of dimensions $d_M$, $d_X$ and $d_Y$ respectively, with a joint covariance matrix given by $\Sigma_{MXY}$. We will make extensive use of the submatrices of $\Sigma_{MXY}$, so we explain their notation here:

- $\Sigma_{XY}$ will denote the $(d_X + d_Y) \times (d_X + d_Y)$ joint (auto-)covariance matrix of the vector $[X^\mathsf{T}, Y^\mathsf{T}]^\mathsf{T}$.
- $\Sigma_{X,Y}$ (note the comma) will denote the $d_X \times d_Y$ cross-covariance matrix between $X$ and $Y$.
- $\Sigma_{XY,M}$ will denote the $(d_X + d_Y) \times d_M$ cross-covariance matrix between the concatenated vector $[X^\mathsf{T}, Y^\mathsf{T}]^\mathsf{T}$ and the vector $M$.

In general, groupings of vectors without commas represent joint covariance, while a comma represents a cross-covariance between the groups on either side of the comma. The same notation will also be used for conditional covariance matrices: for example, $\Sigma_{XY|M}$ is the conditional *joint* covariance of $(X, Y)$ given $M$, while $\Sigma_{X,Y|M}$ is the conditional *cross*-covariance *between* $X$ and $Y$ given $M$.

We will also use an equivalent notation for the joint distribution [12], where $P_{MXY}$ is parameterized as a "broadcast channel" [24, Ch. 15.6] from $M$ to $X$ and $Y$:

$$X = H_X M + N_X \quad \text{and} \quad Y = H_Y M + N_Y. \tag{7}$$

Here, $H_X := \Sigma_{X,M}$ and $H_Y := \Sigma_{Y,M}$ represent *channel gain matrices*, while $N_X$ and $N_Y$ represent additive noise and are not necessarily independent of each other: $[N_X^\mathsf{T}, N_Y^\mathsf{T}]^\mathsf{T} \sim \mathcal{N}(0, \Sigma_{XY|M})$.

**Remark 1.** Without loss of generality, we can assume that $M$, $X$ and $Y$ are all zero-mean, and that $\Sigma_M = I$. Further, we explicitly assume that the $X$ and $Y$ channels are individually whitened, i.e., that $\Sigma_{X|M} = I$ and $\Sigma_{Y|M} = I$. This assumption precludes deterministic relationships between $M$ and $X$ or $Y$, and is required to ensure that information quantities remain finite [12]. $\square$

### 3.2 Optimizing the Union Information

Bertschinger et al. [18] showed that the minimizer for the unique information is also the minimizer for the "union information", $I^\cup(M : X; Y) := UI_X + UI_Y + RI$. In other words, we can also solve the following optimization problem, which yields simpler expressions for the objective and gradient:

$$\widetilde{I^\cup}(M : X; Y) := \min_{Q_{MXY}} I_Q(M; X, Y) \quad \text{s.t.} \quad Q_{MX} = P_{MX}, \; Q_{MY} = P_{MY} \tag{8}$$

Now, suppose $P_{MXY}$ is Gaussian with covariance $\Sigma_{MXY}^P$ and the solution $Q_{MXY}$ is also assumed to be Gaussian with covariance $\Sigma_{MXY}^Q$. Then, the constraint in (8) implies that $\Sigma_{MX}^Q = \Sigma_{MX}^P$ and $\Sigma_{MY}^Q = \Sigma_{MY}^P$. In other words, $\Sigma_M$, $\Sigma_X$, $\Sigma_Y$, and $\Sigma_{M,XY}$ are all constant across $P$ and $Q$. Therefore, the only part of $\Sigma_{MXY}^Q$ that is variable is $\Sigma_{X,Y}^Q$, or equivalently, $\Sigma_{X,Y|M}^Q$.[3] In what follows, we will drop the superscripts denoting the distribution, as this will be clear from context. Generally speaking, we will discuss the optimization problem and thus the distribution will be $Q$.

**Proposition 1.** *The union information for the $\sim_G$-PID of Definition 2 is given by*

$$\widetilde{I_G^\cup} := \min_{\Sigma_{X,Y|M}} \frac{1}{2} \log \det\left(I + \Sigma_M^{-1} \Sigma_{XY,M}^\mathsf{T} \Sigma_{XY|M}^{-1} \Sigma_{XY,M}\right) \quad \text{s.t.} \quad \Sigma_{XY|M} \succcurlyeq 0 \tag{9}$$

---

[3]We can use $\Sigma_{X,Y|M}$ in place of $\Sigma_{X,Y}$ because they differ by a constant: $\Sigma_{X,Y|M} - \Sigma_{X,Y}$ is an off-diagonal block in $\Sigma_{XY} - \Sigma_{XY|M}$, which is equal to $\Sigma_{XY,M} \Sigma_M^{-1} \Sigma_{XY,M}^\mathsf{T}$, which is constant across $P$ and $Q$.

*where the optimization variable $\Sigma_{X,Y|M}$ is an off-diagonal block embedded within $\Sigma_{XY|M}$; all other matrices in the objective are constants that are derived from $\Sigma_{MXY}^P$.*

We solve the above optimization problem using projected gradient descent: we analytically derive the gradient and the projection operator for the constraint set as shown below. Then, we use the RProp algorithm [34, 35] for gradient descent, which independently adjusts the learning rates for each optimization parameter (derivations, and details of implementation and complexity are in App. A). Code for our implementation is available on GitHub [36], and details of the compute configuration are given in App. E.

**Proposition 2.** *The **objective** in Proposition 1 can be simplified to*

$$f(\Sigma_{X,Y|M}) = \frac{1}{2} \log \det \left( I + H_Y^\mathsf{T} H_Y + B^\mathsf{T} S^{-1} B \right), \tag{10}$$

*where $B := (H_X - \Sigma_{X,Y|M} H_Y)$ and $S := (I - \Sigma_{X,Y|M} \Sigma_{X,Y|M}^\mathsf{T})$.*

*The **gradient** of the objective with respect to $\Sigma_{X,Y|M}$ is given by*

$$\nabla f(\Sigma_{X,Y|M}) = S^{-1} B \left( I + H_Y^\mathsf{T} H_Y + B^\mathsf{T} S^{-1} B \right)^{-1} \left( B^\mathsf{T} S^{-1} \Sigma_{X,Y|M} - H_Y^\mathsf{T} \right). \tag{11}$$

*A **projection operator** on to the constraint set $\Sigma_{XY|M} \succcurlyeq 0$ can be obtained as follows: let $\Sigma_{XY|M} =: V \Lambda V^\mathsf{T}$ be the eigenvalue decomposition of $\Sigma_{XY|M}$, with $\Lambda =: \mathrm{diag}(\lambda_i)$. Let $\bar{\lambda}_i := \max(0, \lambda_i)$ represent the rectified eigenvalues, and $\overline{\Lambda} := \mathrm{diag}(\bar{\lambda}_i)$. Then, define*

$$\overline{\Sigma}_{XY|M} := V \overline{\Lambda} V^\mathsf{T}, \tag{12}$$

$$\Sigma_{X,Y|M}^{proj} := \overline{\Sigma}_{X|M}^{-1/2} \overline{\Sigma}_{X,Y|M} \overline{\Sigma}_{Y|M}^{-1/2}, \tag{13}$$

*where $\overline{\Sigma}_{X|M}$, $\overline{\Sigma}_{Y|M}$ and $\overline{\Sigma}_{X,Y|M}$ are submatrices of $\overline{\Sigma}_{XY|M}$.*

## 4 Canonical Gaussian Examples

In this section, we show how well our $\sim_G$-PID estimator performs on a series of Gaussian examples of increasing complexity, which have known ground truth. Barrett [13] showed that, for Gaussian distributions, the $\sim$-PID reduces to the MMI-PID (defined below), whenever $M$ is scalar. These also happen to be cases when the optimal distribution $Q_{MXY}$ is Gaussian [12], and thus the $\sim_G$-PID should recover the ground truth. We then leverage additivity (Property 1) to combine two or more simple examples into complex ones, where ground truth continues to be known.

**Definition 3** (Minimum Mutual Information (MMI) PID). *Let the redundant information be defined as the minimum of the two mutual informations:*

$$RI_{MMI}(M : X; Y) = \min\{I(M; X), I(M; Y)\}. \tag{14}$$

*The remaining MMI-PID components, $UI_{MMI}(M : X \setminus Y)$, $UI_{MMI}(M : Y \setminus X)$ and $SI_{MMI}(M : X; Y)$, follow from equations (1)–(3).*

We first provide a Gaussian analog of Example 1 in Examples 2–4 (for $d_M = d_X = d_Y = 1$). We will use the channel notation described in Equation (7). Complete derivations for these examples (and some nuances that are omitted here) are presented in App. B.

**Example 2** (Pure uniqueness: variable $A$ from Example 1). Suppose $M \sim \mathcal{N}(0, 1)$, $H_X = 1$ and $H_Y = 0$, with $N_X, N_Y \sim$ i.i.d. $\mathcal{N}(0, 1)$. Here, only $X$ receives information about $M$, while $Y$ is pure noise. Thus, $X$ has unique information about $M$ ($UI_X = I(M; X) > 0$), with no unique information in $Y$, and no redundancy or synergy ($UI_Y = RI = SI = 0$). □

**Example 3** (Pure redundancy: variable $B$ from Example 1). Ideally, we would set $M \sim \mathcal{N}(0, 1)$, $X = M$ and $Y = M$. However, for continuous random variables, $I(M; X) = \infty$ when $M = X$. So instead, we set $M \sim \mathcal{N}(0, 1)$, $H_X = 1$ and $H_Y = 1$, with $N_X \sim \mathcal{N}(0, 1)$ while $N_Y = N_X$ (i.e., $X = Y$, so they are both the same noisy version of $M$). In this case, $X$ and $Y$ are fully redundant since they both contain exactly the same information about $M$. Thus, $RI = I(M; (X, Y)) > 0$, while $UI_X = UI_Y = SI = 0$. □

**Example 4** (Pure synergy: variable $C$ from Example 1). We cannot replicate pure synergy for Gaussian variables, but we can approach it in a limit. Let $M \sim \mathcal{N}(0, 1)$, $H_X = 1$ and $H_Y = 0$, with $N_X \sim \mathcal{N}(0, \sigma^2)$ and $N_Y = N_X$ (i.e., $X = M + Y$). Further, let $\sigma^2 \to \infty$. In this case, $I(M; Y) = 0$

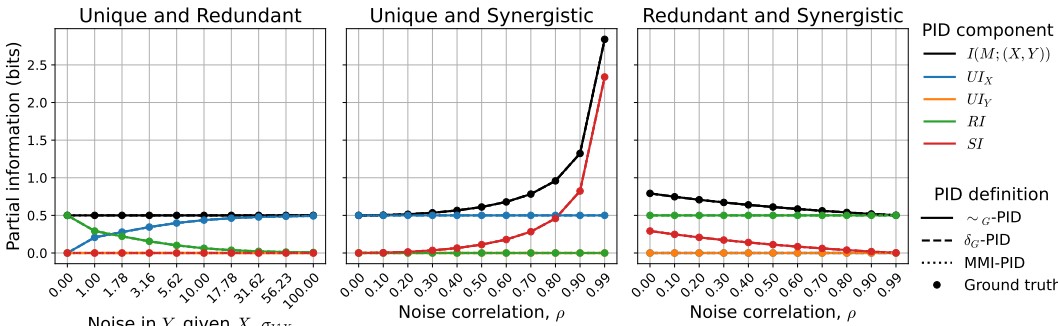

Figure 1: PID values for Examples 5, 6 and 7. The $\sim_G$-PID and the $\delta_G$-PID agree exactly with the MMI-PID, which is known to be the ground truth, since $M$ is scalar [13].

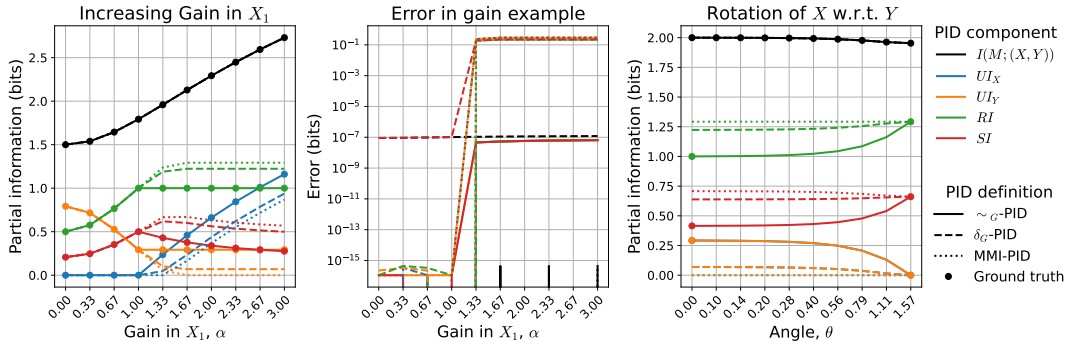

Figure 2: Left and right panels respectively show PID values for Examples 8 and 9, which combine two scalar examples with known ground truth, using Property 1. The middle panel shows the absolute error between each PID definition and the ground truth. The $\sim_G$-PID diverges from the $\delta_G$- and MMI-PIDs, and is the only one that agrees with the ground truth. The $\sim_G$-PID maintains an error less than $10^{-7}$ bits, whereas other definitions have errors greater than 0.1 bits.

and $I(M; X) \to 0$ as $\sigma^2 \to \infty$, so $X$ and $Y$ *individually* convey little to no information about $M$. However, we can recover information about $M$ from $X$ and $Y$ *together* by taking their difference, since $X - Y = M$. Thus, $SI > 0$, while $UI_Y = RI = 0$ and $UI_X \to 0$. ☐

Examples 2, 3 and 4 have been provided solely for intuition. Their PIDs can be inferred directly from Equations (1)–(3). We next describe three one-dimensional examples that each have *two* non-zero PID components. For lack of space, we only provide a brief description and defer details to App. B. We estimate the $\sim_G$-PID (as well as the $\delta_G$-PID [12] and the ground-truth MMI-PID [13]) for these examples and show that all three are equal (see Fig. 1).

**Example 5** (Unique and redundant information). Let $X$ be a noisy representation of $M$, and let $Y$ be a noisy representation of $X$ with standard deviation $\sigma_{Y|X}$. When $Y = X$ (zero noise), this example reduces to Example 3. As $\sigma_{Y|X} \to \infty$, $RI$ reduces while $UI_X$ approaches $I(M; X)$. ☐

**Example 6** (Unique and synergistic information). Let $M \sim \mathcal{N}(0, 1)$, $H_X = 1$, $H_Y = 0$ and $N_X, N_Y \sim \mathcal{N}(0, \sigma^2)$ such that their correlation is $\rho$. When $\sigma^2$ is finite and $\rho = 0$, this example reduces to Example 2, since there can be no synergy between $X$ and $Y$. As $\rho \to 1$, $X - Y \to M$; so the total mutual information $I(M; (X, Y)) \to \infty$, driven by synergy growing unbounded, while the unique component remains constant at $I(M; X)$. ☐

**Example 7** (Redundant and synergistic information). Let $M \sim \mathcal{N}(0, 1)$, $H_X = H_Y = 1$ and $N_X, N_Y \sim \mathcal{N}(0, 1)$ such that their correlation is $\rho$. When $\rho < 1$, $I(M; X)$ and $I(M; Y)$ are both equal by symmetry, and thus equal to $RI$ (see Def. 3 for the MMI-PID, which is ground truth here). As $\rho$ reduces, the two channels $X$ and $Y$ have noisy representations of $M$ with increasingly independent noise terms. Averaging the two, $(X + Y)/2$, will provide more information about $M$ than either one of them individually (i.e., synergy), and thus $SI$ increases as $\rho$ reduces. ☐

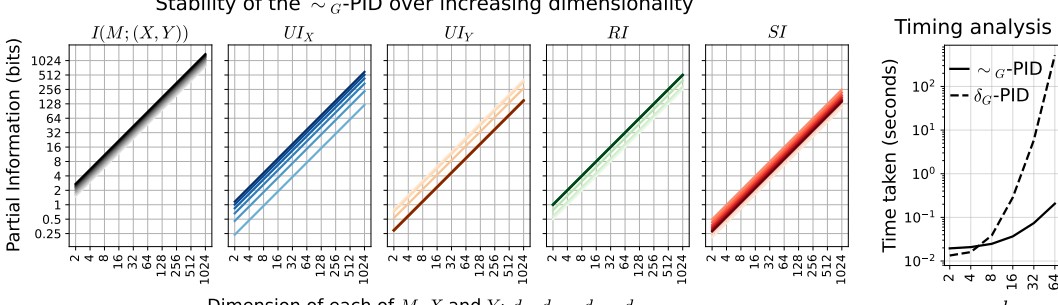

Figure 3: The first five plots on the left show PID values for Example 10. Different shadings represent different values of gain in $X_1$ ($\alpha$) from Example 8. The $\sim_G$-PID doubles every time $d$ doubles as seen by the constant 45° slope on the base-2 log-log plot, even when $d_M = d_X = d_Y = 1024$. The right-most plot shows a timing comparison between the $\sim_G$- and the $\delta_G$-PIDs on this example. The $\delta_G$-PID failed for $d > 64$; the $\sim_G$-PID's timing performance up to $d = 1024$ is shown in App. B.4.

The next set of examples will use the examples presented above in different combinations. This ensures that, where possible, the ground truth remains known in accordance with Property 1. These examples are also designed to reveal the differences between the $\sim$-PID, the MMI-PID and the $\delta$-PID: in particular, they show how the MMI-PID and the $\delta$-PID fail where the $\sim$-PID does not. These examples use two-dimensional $M$, $X$ and $Y$, i.e., $(d_M, d_X, d_Y) = (2, 2, 2)$. A diagrammatic representation of Examples 8 and 9 is given in App. B.2.

**Example 8.** Let $X_1 = \alpha M_1 + N_{X,1}$, $Y_1 = M_1 + N_{Y,1}$, $X_2 = M_2 + N_{X,2}$ and $Y_2 = 3M_2 + N_{Y,2}$, where $M_1, M_2, N_{X,i}, N_{Y,i} \sim$ i.i.d. $\mathcal{N}(0, 1)$, $i = 1, 2$. Here, $(M_1, X_1, Y_1)$ is independent of $(M_2, X_2, Y_2)$, therefore using Property 1, we can add the PID values from their individual decompositions (which each have known ground truth via the MMI-PID since $M_1$ and $M_2$ are scalar). Fig. 2(l) compares the $\sim_G$-PID, the $\delta_G$-PID and the MMI-PID for the joint decomposition of $I(M; (X, Y))$, at different values of $\alpha$, the gain in $X_1$. Only the $\sim_G$-PID matches the ground truth, as it is the only definition here that is additive. □

**Example 9.** Let $M$ and $Y$ be as in Example 8. Suppose $X = H_X R(\theta) M$, where $H_X$ is a diagonal matrix with diagonal entries 3 and 1, and $R(\theta)$ is a $2 \times 2$ rotation matrix that rotates $M$ by an angle $\theta$. When $\theta = 0$, $X$ has higher gain for $M_1$ while $Y$ has higher gain for $M_2$. When $\theta$ increases to $\pi/2$, $X$ and $Y$ have equal gains for both $M_1$ and $M_2$ (barring a difference in sign). Since $(M_1, X_1, Y_1)$ is not independent of $(M_2, X_2, Y_2)$ for all $\theta$, we know the ground truth only at the end-points. Nonetheless, the example shows a difference between the three definitions, as shown in Fig. 2(r). □

**Example 10.** In this example, we test the stability of the $\sim_G$-PID as the dimensionality, $d :=$ $d_M = d_X = d_Y$ increases. By Property 1, if we take two i.i.d. systems of variables $(M, X, Y)$ at dimensionality $d$ and concatenate their respective variables, every PID component of the composite system of dimensionality $2d$ should be double that of the original. This process can be repeated, taking two independent $2d$-dimensional systems and concatenating them to create a $4d$-dimensional system. Fig. 3 shows precisely this process starting with the system in Example 8 with $d = 2$, and continually doubling its size until $d = 1024$. The $\sim_G$-PID accurately matches ground truth by doubling in value, and remains stable with small relative errors (shown in App. B.5). The $\sim_G$-PID also runs much faster than the $\delta_G$-PID on this example, providing a speed-up of more than $1000\times$ at $d = 64$ (see right-most plot in Fig. 3, and extended results in App. B.4). □

**Remark 2.** In the above examples, the ground truth referred to the $\sim$-PID of Def. 1, since it was inferred using Def. 3 and additivity (Property 1). Since our method for computing the $\sim_G$-PID recovers the ground truth $\sim$-PID, we can infer that the distribution of the optimal $Q_{MXY}$ for the $\sim$-PID is in fact Gaussian in these examples. This provides empirical evidence in support of the conjecture stated in Sec. 3. □

## 5    Estimation and Bias-correction for the $\sim_G$-PID

Having discussed how to compute the $\sim_G$-PID and shown that it agrees well with ground truth in several canonical examples, we discuss how the $\sim_G$-PID may be estimated from data. Given a sample

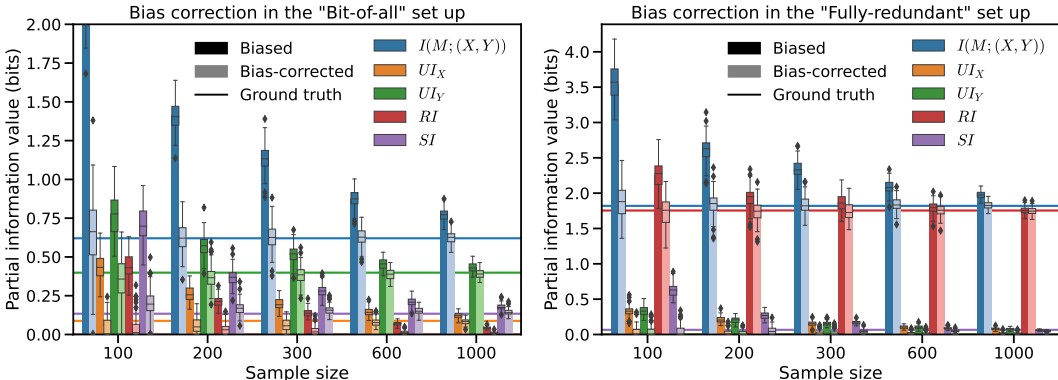

Figure 4: Empirical evaluation of bias-corrected PID estimates with increasing sample size for two configurations with $d_M = d_X = d_Y = 10$, as described in Section 5. Solid horizontal lines represent ground truth (as computed from the true covariance matrix); dark colored bars represent biased PID components and light colored bars represent bias-corrected PID components, as estimated from the sample covariance matrix. Overlaid box plots indicate results from 100 random draws. Empirically, we find our bias-corrected estimates are both unbiased and consistent.

of $n$ realizations of $M$, $X$ and $Y$ drawn from $P_{MXY}$, we may estimate the sample joint covariance matrix $\hat{\Sigma}_{MXY}$. We therefore use the straightforward "plug-in" estimator for the $\sim_G$-PID, by using $\hat{\Sigma}_{MXY}$ in place of $\Sigma_{MXY}$ in the optimization problem in equation (9).

However, it is well-known that estimators of information-theoretic quantities suffer from large biases for moderate sample sizes [37]. Cai et al. [38] characterized the bias in the entropy of a $d$-dimensional Gaussian random vector, for a fixed sample size $n$.

**Proposition 3** (Bias in Gaussian entropy [38]). *Suppose $M \in \mathbb{R}^{d_M}$ has an auto-covariance matrix $\Sigma_M$. The entropy of $M$ is $H(M) = \frac{1}{2}\log\det(2\pi e\Sigma_M)$ when $\Sigma_M$ is known [24]. For the sample covariance matrix $\hat{\Sigma}_M$, the bias is given by:*

$$\text{Bias}\big[\hat{H}(M)\big] = \sum_{k=1}^{d_M} \log(1 - k/n). \tag{15}$$

For a proof, we refer the reader to [38, Corollary 2]. This result may be naturally extended to compute the bias of each of the mutual information quantities in the LHS of equations (1)–(3):

**Corollary 4** (Bias in Gaussian mutual information). *For the joint mutual information $I(M;(X,Y))$,*

$$\text{Bias}\big[\hat{I}\big(M;(X,Y)\big)\big] = \sum_{k=1}^{d_M} \log(1 - k/n) + \sum_{k=1}^{d_X+d_Y} \log(1 - k/n) - \sum_{k=1}^{d_M+d_X+d_Y} \log(1 - k/n) \tag{16}$$

This follows directly from the fact that $I(M;(X,Y)) = H(M) + H(X,Y) - H(M,X,Y)$. Similarly, we can compute the bias of $\hat{I}(M;X)$ and $\hat{I}(M;Y)$. But this does not uniquely determine the bias in the individual PID components, and as with defining PIDs, we need to decompose the bias in Corollary 4 across the four PID components such that they agree with these constraints. We solve this problem by defining a bias-corrected version of the union information from Proposition 1.

**Definition 4** (Bias-corrected Union Information). *We assign the bias in the union information to be the* same fraction *as the bias in the joint mutual information $I(M;(X,Y))$. This gives rise to a bias-corrected estimate of the union information:*

$$\widetilde{I_G^\cup}\Big|_{\text{bias-corr}}(M:X;Y) := \widetilde{I_G^\cup}(M:X;Y)\left(1 - \frac{\text{Bias}\big[\hat{I}\big(M;(X,Y)\big)\big]}{\hat{I}\big(M;(X,Y)\big)}\right). \tag{17}$$

We do not analyze theoretically whether the bias-corrected union information is consistent and unbiased, thus, it may still have some residual bias relative to the true union information. However, we find empirically that this bias correction process works reasonably well and appears both consistent and unbiased, in a number of examples. Figure 4 shows biased and bias-corrected PID values for

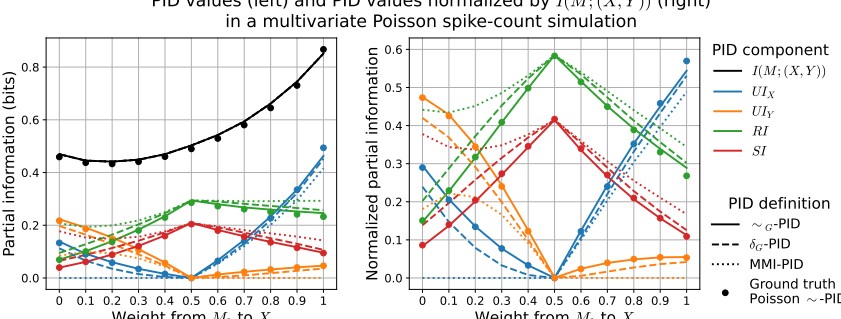

Figure 5: A comparison of the $\sim_G$-PID, the $\delta_G$-PID and the MMI-PID for a multivariate Poisson system. The ground truth is a discrete $\sim$-PID computed using the package of Banerjee et al. [30]. The $\sim_G$-PID comes closest to the ground truth (possibly because they compute the same PID definition), despite the fact that the $\sim_G$-PID only uses the covariance matrix of the Poisson distribution, whereas the ground truth uses knowledge of the distribution itself.

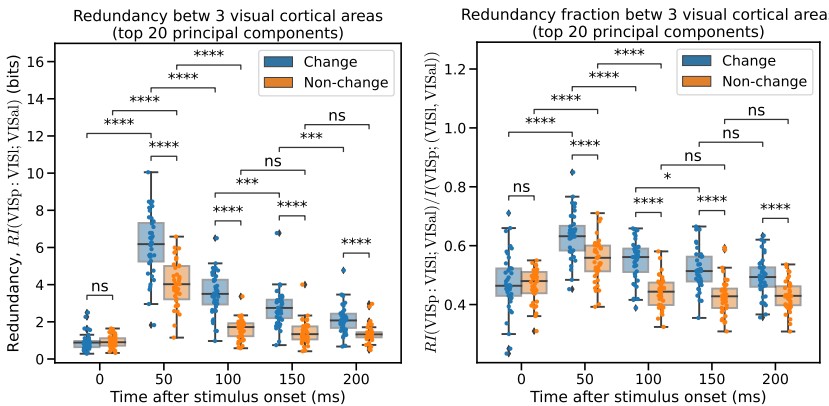

Figure 6: Bias-corrected redundancy estimates for information about VISp activity that is shared between VISl and VISal: redundancy in bits (left) and redundancy as a fraction of total mutual information (right). Data spread is across 40 mice. Statistical comparisons use a two-sided Mann-Whitney-Wilcoxon test. Observe that there is greater and more sustained redundancy on flashes corresponding to image changes, which are behaviorally relevant and linked to rewards in this task.

100 runs of two configurations called "Bit-of-all" (with a little bit of each PID component) and "Fully-redundant" (which has predominantly redundancy), with $d_M = d_X = d_Y = 10$ (details and additional setups in App. C). We find that bias correction brings the PID values closer to their true values even at small sample sizes. In App. C.3, we also include a preliminary analysis of the variance of PID estimates using bootstrap [39, Ch. 8].

## 6 Application to Simulated and Real Neural Data

So far, we have only considered the $\sim_G$-PID when applied to *Gaussian* $P_{MXY}$. Although Def. 2 strictly applies only to Gaussian $P_{MXY}$, the estimation process in Sec. 5 relies only on a sample covariance matrix, which is well-defined for a wide variety of non-Gaussian distributions. Many applications where PIDs could be useful have non-Gaussian data. For instance, there is great interest in applying PIDs in neuroscience (e.g., to understand how multiple brain regions jointly encode and communicate information [6, 40]), but spike-count distributions are non-Gaussian.

**Simulated neural data.** To show that our $\sim_G$-PID estimates provide reasonable results on non-Gaussian spiking neural data, we first simulate spike-count data using Poisson random variables (following [12]; described in App. D.1). We evaluate the ground truth $\sim$-PID for this distribution using the discrete PID estimator of Banerjee et al. [20]. The $\sim_G$-PID is estimated from a sample covariance matrix using $10^6$ realizations of $M$, $X$ and $Y$. We find that the $\sim_G$-PID closely matches

the ground truth for a range of parameter values, despite the fact that the $\sim_G$-PID is effectively computed on a Gaussian approximation of a Poisson distribution (Fig. 5). More examples of the $\sim_G$-PID applied to non-Gaussian data are provided in App. D.2 (also see App. A.2). We conclude that it is reasonable to use and interpret the $\sim_G$-PID on non-Gaussian spike count data.

**Real neural data.** We then applied our bias-corrected $\sim_G$-PID estimator to the Visual Behavior Neuropixels dataset collected by us at the Allen Institute [41]. We recorded over 80 mice using six neuropixels probes targeting various regions of visual cortex, while the mice were engaged in a visual change-detection task. In the task, images from a set of 8 natural scenes were presented in 250 ms flashes, at intervals of 750 ms; the image would stay the same for a variable number of flashes after which it would change to a new image. The mouse had to lick to receive a water reward when the image changed. Thus, a given image flash could be a behaviorally relevant target if the previous image was different, or not, if the previous image was the same.

We used our bias-corrected PID estimator to understand how information is processed along the visual hierarchy during this task. We estimated the $\sim_G$-PID to understand how information contained in the spiking activity of primary visual cortex (VISp) was represented in two higher-order visual cortical areas, VISl and VISal. We aligned trials to the onset of a stimulus flash, binned spikes in 50 ms intervals and considered the top 20 principal components (to achieve reasonable estimates at these sample sizes; explained further in App. D.5) from each region in each time bin. We computed the $\sim_G$-PID on the sample covariance matrix of these principal components (shown in Fig. 6). We found that there was a significantly larger amount of redundant information about VISp activity between VISl and VISal for stimulus flashes corresponding to an image change, compared to flashes that were not changes (Fig. 6(l)). The larger redundancy was also sustained slightly longer for flashes corresponding to changes, than non-change flashes. Both of these effects were maintained even when the redundancy was normalized by the joint mutual information, suggesting that the effect was not purely due to an increase in the total amount of information (Fig. 6(r)). Our results suggest that the visual cortex propagates information throughout the hierarchy more robustly when such information is relevant for behavior.

## 7 Discussion

In this paper, we proposed a new and efficient method for estimating the $\sim_G$-PID for Gaussian distributions. We showed that our method recovers the ground truth and suitably corrects for bias through a series of examples. In particular, Fig. 4 showed how large the biases can be at small sample sizes, which makes bias correction particularly important in neuroscientific settings where sample sizes are often small.

We focused on Gaussian distributions, as they have historically been a starting point for many estimators (e.g., correlation is used as a measure of dependence, but zero correlation implies independence only in the Gaussian case). We were able to show ground-truth validation for our $\sim_G$-PID estimator at high dimensionalities only thanks to the existence of closed-form results on scalar Gaussians.

While our central claims and results applied to Gaussian distributions, our method for computing PIDs did not immediately break down for distributions close to Gaussian in some limit (e.g., Poisson). The effective spiking rate used in our multivariate Poisson distribution was also not particularly large (see App. D.1); we would expect our estimates to improve for higher firing rates, since the Poisson distribution will then be more Gaussian.

**Limitations.** Our work has several limitations that require further theory and simulations to resolve, the most important of which are: (1) Our estimator is technically a bound on the PID values because we assume Gaussian optimality in Definition 2; (2) Our bias-correction method is heuristic: we do not provide a rigorous theoretical characterization of the bias of PID values.

**Broader impacts.** Our work is mainly methodological, so the scope for negative impacts depends on how the methods might be used. For example, incorrect interpretations drawn from the use of our PID estimators may affect scientific conclusions. In particular, the PID is inherently a correlational quantity and carries the same caveats: it may not be appropriate to make causal interpretations on the basis of observed PID values. Also, despite our best efforts to explore a variety of systems, we cannot tell how accurate our bias-correction method will be in novel configurations.

## Acknowledgments and Disclosure of Funding

We thank Łukasz Kuśmierz for providing a valuable reference on the bias of Shannon entropy estimates. We also thank Gabe Schamberg and Christof Koch for helpful discussions.

We wish to thank the Allen Institute founder, Paul G. Allen, for his vision, encouragement, and support. Praveen Venkatesh was supported by a Shanahan Family Foundation Fellowship at the Interface of Data and Neuroscience, supported in part by the Allen Institute. Stefan Mihalas was in part supported by NSF 2223725, NIH R01EB029813, and RF1DA055669 grants.

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

# Gaussian Partial Information Decomposition:
# Bias Correction and Application to High-dimensional Data

# Appendices

Praveen Venkatesh, Corbett Bennett, Sam Gale, Tamina K. Ramirez, Greggory Heller,
Séverine Durand, Shawn Olsen, Stefan Mihalas

## A   Supplementary Material for Sections 2 and 3

### A.1   Implications of the Additivity Property

As mentioned in the main text, additivity states that the PID values of an isolated system should not depend on the PID values of another isolated system. Without additivity, it is not possible to examine independent systems in isolation, since broadening your view to include a different isolated system could change the PID values of the first system.

Hypothetically, suppose we have two separate individuals (labeled 1 and 2) receiving completely independent stimuli, and we examine the PID between the activity in brain regions $M$, $X$ and $Y$ in each of their brains. Then the *total* unique information that $X_1$ and $X_2$ have about $M_1$ and $M_2$ with respect to $Y_1$ and $Y_2$ should be equal to the *sum* of the unique information in each of the individuals taken separately.

As another example, suppose we are trying to examine visual information flow and auditory information flow in a multi-sensory integration task. We may want to understand the degree to which the activity in the sub-regions of the visual and auditory systems depend on each other. A reasonable null model of independence between the two systems would be that the PID values of the joint system will be equal to the sum of the PID values in the two individual systems. Then, measuring the actual degree to which the joint PID value is not equal to the sum will be a meaningful measure of dependence between the systems. This would only be possible with a PID definition that guaranteed additivity of independent sub-systems. However, since the $\delta$-PID does not satisfy the additivity (Property 1), we cannot guarantee that the aforementioned null would be the correct null model, and we would not be able to perform such an analysis.

### A.2   Explaining the Exponential Reduction in the Number of Optimization Variables

In Def. 2, we restrict $Q_{MXY}$ to be jointly Gaussian. This reduces the number of optimization variables from being exponential in the dimensionality to quadratic, as we show here.

Previous discrete $\sim$-PID estimators [20, 21] have found efficient methods to solve the optimization problem of Def. 1 without reducing the number of optimization variables. If $P_{MXY}$ is discrete in Def. 1, and each dimension of $M$, $X$ and $Y$ has a support of size $K$ (i.e., $M_i$, $X_i$ and $Y_i$ can each take one of $K$ discrete values), then the total number of degrees of freedom in $\Delta_P$ is $\mathcal{O}(K^{(d_M+d_X+d_Y)})$. This corresponds to the total support of a discrete $Q_{MXY}$, which can be an arbitrarily complex discrete distribution.

In contrast, the $\sim_G$-PID in Def. 2 reduces the dimensionality of the optimization space by assuming that $Q_{MXY}$ is Gaussian. $Q_{MXY}$ is then completely parameterized by its covariance matrix, $\Sigma^Q_{MXY}$. In fact, given the additional constraints in Def. 2, the only part of $\Sigma^Q_{MXY}$ that is variable is $\Sigma^Q_{X,Y}$, as explained in Sec. 3.2. This matrix has only $d_X \times d_Y$ variables. Thus, by restricting the optimization space to jointly Gaussian distributions, we have reduced the number of variables from exponential in the dimensionality to quadratic.

As a concrete example, consider a neuroscientific setting (as in Sec. 6) where $M$, $X$ and $Y$ represent the spiking activity of neurons in three different brain regions, so that $d_M$, $d_X$ and $d_Y$ represent the number of neurons, and $K$ represents the maximum number of spikes. Makkeh et al. [21] demonstrated their method for a maximum support size of $K = 18$ with $d_M = d_X = d_Y = 1$, for a total support of approximately $18^3 - 1 = 5831$ in $Q_{MXY}$. If we assume a single neuron can have at most $K = 10$ spikes, then we get an effective dimensionality of $\log_{10}(5831) < 4$. In other words,

these discrete $\sim$-PID estimators can handle around 4 neurons across all three regions $M$, $X$ and $Y$ (which is also what we use in the simulation of Fig. 5 in Sec. 6), whereas our $\sim_G$-PID estimator can handle several hundreds if not thousands of neurons (as seen in Example 10).

### A.3 Proofs of Propositions 1 and 2

*Proof of Proposition 1.* Firstly, the differential entropy of a Gaussian random variable $M$ with covariance matrix $\Sigma_M$ is given by [43, Thm. 8.4.1]:

$$h(M) = \frac{1}{2} \log \det(2\pi e \Sigma_M). \tag{18}$$

Secondly, for a joint Gaussian distribution $P_{MXY}$ parameterized by a covariance matrix $\Sigma_{MXY}$, the conditional covariance matrix can be written as [44, Sec. 8.1.3]:

$$\Sigma_{XY|M} = \Sigma_{XY} - \Sigma_{XY,M} \Sigma_M^{-1} \Sigma_{XY,M}^\mathsf{T} \tag{19}$$

$$\Rightarrow \quad \Sigma_{XY} = \Sigma_{XY|M} + \Sigma_{XY,M} \Sigma_M^{-1} \Sigma_{XY,M}^\mathsf{T} \tag{20}$$

Using these two equations, we can derive the mutual information between $M$ and $(X, Y)$ as follows:

$$I(M; (X, Y)) = h(X, Y) - h(X, Y \mid M) \tag{21}$$

$$\overset{(a)}{=} \frac{1}{2} \log \det(2\pi e \Sigma_{XY}) - \frac{1}{2} \log \det(2\pi e \Sigma_{XY|M}) \tag{22}$$

$$= \frac{1}{2} \log\big((2\pi e)^{d_M} \det(\Sigma_{XY})\big) - \frac{1}{2} \log\big((2\pi e)^{d_M} \det(\Sigma_{XY|M})\big) \tag{23}$$

$$= \frac{1}{2} \log\left(\frac{\det(\Sigma_{XY})}{\det(\Sigma_{XY|M})}\right) \tag{24}$$

$$\overset{(b)}{=} \frac{1}{2} \log\left(\frac{\det(\Sigma_{XY|M} + \Sigma_{XY,M} \Sigma_M^{-1} \Sigma_{XY,M}^\mathsf{T})}{\det(\Sigma_{XY|M})}\right) \tag{25}$$

$$= \frac{1}{2} \log\left(\frac{\det(\Sigma_{XY|M}) \det(I + \Sigma_{XY|M}^{-1} \Sigma_{XY,M} \Sigma_M^{-1} \Sigma_{XY,M}^\mathsf{T})}{\det(\Sigma_{XY|M})}\right) \tag{26}$$

$$= \frac{1}{2} \log \det(I + \Sigma_{XY|M}^{-1} \Sigma_{XY,M} \Sigma_M^{-1} \Sigma_{XY,M}^\mathsf{T}) \tag{27}$$

$$\overset{(c)}{=} \frac{1}{2} \log \det(I + \Sigma_M^{-1} \Sigma_{XY,M}^\mathsf{T} \Sigma_{XY|M}^{-1} \Sigma_{XY,M}), \tag{28}$$

where in (a) we used equation (18), in (b) we used equation (20), and in (c) we used the fact that $\det(I + AB) = \det(I + BA)$.

The remainder of the proof follows from the arguments presented below equation (8). The constraint in Proposition 1 arises because, when optimizing over $\Sigma_{X,Y|M}$, we require $\Sigma_{MXY}$ to be a valid positive semidefinite covariance matrix, i.e., $\Sigma_{MXY} \succcurlyeq 0$. This happens if and only if $\Sigma_M$ and its Schur complement in $\Sigma_{MXY}$ are both positive semidefinite, i.e., $\Sigma_M \succcurlyeq 0$ and $\Sigma_M - \Sigma_{M,XY} \Sigma_{XY}^{-1} \Sigma_{M,XY}^\mathsf{T} = \Sigma_{XY|M} \succcurlyeq 0$. $\qquad\square$

*Proof of Proposition 2.* The proof is divided into three parts consisting of derivations for the objective, the gradient and the projection operator.

**Objective.** After whitening the $P_{X|M}$ and the $P_{Y|M}$ channels, and assuming that $\Sigma_M = I$ (see Remark 1), without loss of generality we have that

$$\Sigma_{X,M} = \mathbb{E}\big[(H_X M + N_X) M^\mathsf{T}\big] = H_X \mathbb{E}[MM^\mathsf{T}] = H_X \tag{29}$$

$$\Sigma_{XY|M} = \begin{bmatrix} I & \Sigma_{X,Y|M} \\ \Sigma_{X,Y|M}^\mathsf{T} & I \end{bmatrix} \tag{30}$$

$$\Rightarrow \quad \Sigma_M^{-1} \Sigma_{XY,M}^\mathsf{T} \Sigma_{XY|M}^{-1} \Sigma_{XY,M} = \begin{bmatrix} H_X^\mathsf{T} & H_Y^\mathsf{T} \end{bmatrix} \begin{bmatrix} I & \Sigma_{X,Y|M} \\ \Sigma_{X,Y|M}^\mathsf{T} & I \end{bmatrix}^{-1} \begin{bmatrix} H_X \\ H_Y \end{bmatrix} \tag{31}$$

For the sake of brevity, let $\Sigma$ represent the optimization variable $\Sigma_{X,Y|M}$, and let $S$ be its Schur complement in $\Sigma_{XY|M}$, $I - \Sigma\Sigma^{\mathsf{T}}$. Then, the inverse of $\Sigma_{XY|M}$ can be written as [44, Sec. 9.1.5]:

$$\Sigma_{XY|M}^{-1} = \begin{bmatrix} S^{-1} & -S^{-1}\Sigma \\ -\Sigma^{\mathsf{T}}S^{-1} & I + \Sigma^{\mathsf{T}}S^{-1}\Sigma \end{bmatrix} \tag{32}$$

Therefore, we get:

$$\Sigma_M^{-1}\Sigma_{XY,M}^{\mathsf{T}}\Sigma_{XY|M}^{-1}\Sigma_{XY,M} = \begin{bmatrix} H_X^{\mathsf{T}} & H_Y^{\mathsf{T}} \end{bmatrix} \begin{bmatrix} S^{-1} & -S^{-1}\Sigma \\ -\Sigma^{\mathsf{T}}S^{-1} & I + \Sigma^{\mathsf{T}}S^{-1}\Sigma \end{bmatrix} \begin{bmatrix} H_X \\ H_Y \end{bmatrix} \tag{33}$$

$$= H_Y^{\mathsf{T}}H_Y + (H_X - \Sigma H_Y)^{\mathsf{T}}S^{-1}(H_X - \Sigma H_Y) \tag{34}$$

Thus, setting $B := H_X - \Sigma H_Y$, the optimization problem in Proposition 1 reduces to

$$\min_{\Sigma} \quad \frac{1}{2}\log\det\left(I + H_Y^{\mathsf{T}}H_Y + B^{\mathsf{T}}S^{-1}B\right)$$
$$\text{s.t.} \quad \Sigma_{XY|M} \succcurlyeq 0 \tag{35}$$

**Gradient.** Let the objective derived in the previous section be called $f(\Sigma)$, where $\Sigma := \Sigma_{X,Y|M}$ as before. We can compute the gradient of $f$ with respect to $\Sigma$ using standard identities from matrix calculus. First, note that the gradient of a scalar function with respect to a matrix is itself a matrix with entries as follows:

$$\nabla f(\Sigma)\Big|_{ij} = \frac{\partial f}{\partial \Sigma_{ij}}(\Sigma). \tag{36}$$

Considering each element of this matrix:

$$\frac{\partial f}{\partial \Sigma_{ij}}(\Sigma) = \frac{1}{2}\frac{\partial}{\partial \Sigma_{ij}}\log\det(I + H_Y^{\mathsf{T}}H_Y + B^{\mathsf{T}}S^{-1}B)\Big|_{\Sigma} \tag{37}$$

$$\overset{(a)}{=} \frac{1}{2}\mathrm{Tr}\left\{(I + H_Y^{\mathsf{T}}H_Y + B^{\mathsf{T}}S^{-1}B)^{-1}\frac{\partial}{\partial \Sigma_{ij}}(I + H_Y^{\mathsf{T}}H_Y + B^{\mathsf{T}}S^{-1}B)\right\}\Big|_{\Sigma} \tag{38}$$

$$\overset{(b)}{=} \frac{1}{2}\mathrm{Tr}\left\{(I + H_Y^{\mathsf{T}}H_Y + B^{\mathsf{T}}S^{-1}B)^{-1}\frac{\partial}{\partial \Sigma_{ij}}(B^{\mathsf{T}}S^{-1}B)\right\}\Big|_{\Sigma}, \tag{39}$$

where in (a), we have used the identity $\partial\log\det(X) = \mathrm{Tr}\{X^{-1}\partial(X)\}$ [44, Sec. 2], while in (b), we use the fact that only $B$ and $S$ depend on $\Sigma$ implicitly, with the other terms being constants.

Expanding the partial derivative alone, we get:

$$\frac{\partial}{\partial \Sigma_{ij}}(B^{\mathsf{T}}S^{-1}B)\Big|_{\Sigma} = \left[\frac{\partial}{\partial \Sigma_{ij}}(B^{\mathsf{T}})\cdot S^{-1}B + B^{\mathsf{T}}\cdot\frac{\partial}{\partial \Sigma_{ij}}(S^{-1})\cdot B + B^{\mathsf{T}}S^{-1}\cdot\frac{\partial}{\partial \Sigma_{ij}}(B)\right]_{\Sigma}, \tag{40}$$

wherein

$$\frac{\partial}{\partial \Sigma_{ij}}(B)\Big|_{\Sigma} = \frac{\partial}{\partial \Sigma_{ij}}(H_X - \Sigma H_Y)\Big|_{\Sigma} \tag{41}$$

$$\overset{(b)}{=} -J^{ij}H_Y, \tag{42}$$

$$\frac{\partial}{\partial \Sigma_{ij}}(S^{-1})\Big|_{\Sigma} \overset{(c)}{=} -S^{-1}\frac{\partial S}{\partial \Sigma_{ij}}S^{-1}\Big|_{\Sigma} \tag{43}$$

$$= -S^{-1}\frac{\partial}{\partial \Sigma_{ij}}(I - \Sigma\Sigma^{\mathsf{T}})S^{-1}\Big|_{\Sigma} \tag{44}$$

$$\overset{(d)}{=} -S^{-1}(-J^{ij}\Sigma^{\mathsf{T}} - \Sigma J^{ij\mathsf{T}})S^{-1}, \tag{45}$$

where $J^{ij}$ is the *single-entry matrix*, containing a 1 at location $(i, j)$ and 0's everywhere else; in (b) and (d), we use the fact that $\partial X/\partial X_{ij} = J^{ij}$ [44, Sec. 9.7.6]; and in (c) we use the identity

$\partial(X^{-1}) = X^{-1}\partial(X)X^{-1}$. Therefore, (40) becomes

$$\frac{\partial}{\partial\Sigma_{ij}}\big(B^\mathsf{T}S^{-1}B\big)\Big|_\Sigma \tag{46}$$

$$= -(J^{ij}H_Y)^\mathsf{T}S^{-1}B \;+\; B^\mathsf{T}S^{-1}(J^{ij}\Sigma^\mathsf{T} + \Sigma J^{ij\mathsf{T}})S^{-1}B \;+\; B^\mathsf{T}S^{-1}(-J^{ij}H_Y) \tag{47}$$

$$= -H_Y^\mathsf{T}J^{ij\mathsf{T}}S^{-1}B \;+\; B^\mathsf{T}S^{-1}J^{ij}\Sigma^\mathsf{T}S^{-1}B \;+\; B^\mathsf{T}S^{-1}\Sigma J^{ij\mathsf{T}}S^{-1}B \;-\; B^\mathsf{T}S^{-1}J^{ij}H_Y. \tag{48}$$

Putting it all together, and letting $A := (I + H_Y^\mathsf{T}H_Y + B^\mathsf{T}S^{-1}B)$, (39) becomes

$$\frac{\partial f}{\partial\Sigma_{ij}}(\Sigma) = \frac{1}{2}\mathrm{Tr}\Big\{A^{-1}\big(-H_Y^\mathsf{T}J^{ij\mathsf{T}}S^{-1}B \;+\; B^\mathsf{T}S^{-1}J^{ij}\Sigma^\mathsf{T}S^{-1}B$$
$$+\; B^\mathsf{T}S^{-1}\Sigma J^{ij\mathsf{T}}S^{-1}B \;-\; B^\mathsf{T}S^{-1}J^{ij}H_Y\big)\Big\} \tag{49}$$

$$= \frac{1}{2}\Big[-\mathrm{Tr}\big\{A^{-1}H_Y^\mathsf{T}J^{ij\mathsf{T}}S^{-1}B\big\} \;+\; \mathrm{Tr}\big\{A^{-1}B^\mathsf{T}S^{-1}J^{ij}\Sigma^\mathsf{T}S^{-1}B\big\}$$
$$+\; \mathrm{Tr}\big\{A^{-1}B^\mathsf{T}S^{-1}\Sigma J^{ij\mathsf{T}}S^{-1}B\big\} \;-\; \mathrm{Tr}\big\{A^{-1}B^\mathsf{T}S^{-1}J^{ij}H_Y\big\}\Big] \tag{50}$$

$$\overset{(e)}{=} \frac{1}{2}\Big[-\mathrm{Tr}\big\{S^{-1}BA^{-1}H_Y^\mathsf{T}J^{ij\mathsf{T}}\big\} \;+\; \mathrm{Tr}\big\{\Sigma^\mathsf{T}S^{-1}BA^{-1}B^\mathsf{T}S^{-1}J^{ij}\big\}$$
$$+\; \mathrm{Tr}\big\{S^{-1}BA^{-1}B^\mathsf{T}S^{-1}\Sigma J^{ij\mathsf{T}}\big\} \;-\; \mathrm{Tr}\big\{H_Y A^{-1}B^\mathsf{T}S^{-1}J^{ij}\big\}\Big], \tag{51}$$

where in (e), we have used the fact that the trace of a matrix product is invariant under cyclic permutations of the matrices within the product.

Finally, using the fact that $\mathrm{Tr}\{W^\mathsf{T}J^{ij}\} = \mathrm{Tr}\{WJ^{ij\mathsf{T}}\} = W_{ij}$ for any matrix $W$ [44, Sec. 9.7.5],

$$\frac{\partial f}{\partial\Sigma_{ij}}(\Sigma) = \frac{1}{2}\Big[-2\big(S^{-1}BA^{-1}H_Y^\mathsf{T}\big)_{ij} \;+\; 2\big(S^{-1}BA^{-1}B^\mathsf{T}S^{-1}\Sigma\big)_{ij}\Big] \tag{52}$$

$$= \Big[S^{-1}BA^{-1}\big(B^\mathsf{T}S^{-1}\Sigma - H_Y^\mathsf{T}\big)\Big]_{ij} \tag{53}$$

$$\Rightarrow \quad \nabla f(\Sigma) = S^{-1}BA^{-1}\big(B^\mathsf{T}S^{-1}\Sigma - H_Y^\mathsf{T}\big) \tag{54}$$

$$= S^{-1}B\big(I + H_Y^\mathsf{T}H_Y + B^\mathsf{T}S^{-1}B\big)^{-1}\big(B^\mathsf{T}S^{-1}\Sigma - H_Y^\mathsf{T}\big). \tag{55}$$

**Projection operator.** Recall that the optimization variable, $\Sigma := \Sigma_{X,Y|M}$ is an off-diagonal block of $\Sigma_{XY|M}$, which is the matrix upon which the constraint is defined:

$$\Sigma_{XY|M} = \begin{bmatrix} I & \Sigma \\ \Sigma^\mathsf{T} & I \end{bmatrix}, \tag{56}$$

wherein the diagonal blocks are identity due to Remark 1. For the purposes of this section, let us suppose $\Sigma_{XY|M}$ is a function of $\Sigma$, $\Sigma_{XY|M} =: g(\Sigma)$, so that the constraint may be written as $g(\Sigma) \succeq 0$. A suitable projection operator, therefore, will accept a value $\Sigma_0$ (that may violate $g(\Sigma_0) \succeq 0$) and find a point $\Sigma^{proj}$ close to it that satisfies the constraint, i.e., $g(\Sigma^{proj}) \succeq 0$.

We do not find the "orthogonal" projection operator, which has the minimum distance $\|\Sigma^{proj} - \Sigma_0\|$ in some norm. Instead, we propose a simple heuristic to find a $\Sigma^{proj}$ which satisfies the constraint.

If $\Sigma_0$ satisfies the constraint, then we are done, so let us assume that $g(\Sigma_0) \nsucceq 0$. Then, we can find a matrix $\overline{\Sigma}_{XY|M}$ which is close to $g(\Sigma_0)$ and satisfies $\overline{\Sigma}_{XY|M} \succeq 0$ as follows: let the eigenvalue decomposition of $g(\Sigma_0)$ be given by $V\Lambda V^\mathsf{T}$, with $\Lambda =: \mathrm{diag}(\lambda_i)$ being the diagonal matrix consisting of its eigenvalues $\lambda_i$. Then, since $g(\Sigma_0)$ is not positive semidefinite, $\exists\, i$ s.t. $\lambda_i < 0$. We set $\overline{\lambda}_i := 0$ for all such $i$; effectively, $\overline{\lambda}_i = \max\{0, \lambda_i\} \,\forall\, i$. We then reconstruct the matrix using these "rectified" eigenvalues and set it to be $\overline{\Sigma}_{XY|M} := V\overline{\Lambda}V^\mathsf{T}$, where $\overline{\Lambda} = \mathrm{diag}(\overline{\lambda}_i)$.

Now, we need to find $\Sigma$ such that $g(\Sigma) = \overline{\Sigma}_{XY|M}$. However, $\overline{\Sigma}_{XY|M}$ may not have identity matrices on its diagonal blocks, i.e., it might not correspond to a whitened channel. We therefore whiten

$\overline{\Sigma}_{XY|M}$ as follows:

$$\overline{\Sigma}_{XY|M}^{whitened} = \begin{bmatrix} \overline{\Sigma}_{X|M}^{-1/2} & 0 \\ 0 & \overline{\Sigma}_{Y|M}^{-1/2} \end{bmatrix} \overline{\Sigma}_{XY|M} \begin{bmatrix} \overline{\Sigma}_{X|M}^{-1/2} & 0 \\ 0 & \overline{\Sigma}_{Y|M}^{-1/2} \end{bmatrix}, \tag{57}$$

where $\overline{\Sigma}_{X|M}$ and $\overline{\Sigma}_{Y|M}$ are the diagonal blocks of $\overline{\Sigma}_{XY|M}$. Crucially, since the matrix multiplying $\overline{\Sigma}_{XY|M}$ on either side is itself (the inverse square-root of) a covariance matrix (and hence positive semidefinite), $\overline{\Sigma}_{XY|M}^{whitened}$ is also positive semidefinite.

Now, the off-diagonal block of $\overline{\Sigma}_{XY|M}^{whitened}$ will satisfy $g(\cdot) = \overline{\Sigma}_{XY|M}^{whitened} \succcurlyeq 0$. This off-diagonal block forms the output of our projection operation and can be written as

$$\Sigma_{X,Y|M}^{proj} = \overline{\Sigma}_{X|M}^{-1/2} \overline{\Sigma}_{XY|M} \overline{\Sigma}_{Y|M}^{-1/2}, \tag{58}$$

which comes directly from equation (57). $\qquad\square$

## A.4 Details of $\sim_G$-PID Optimization and RProp Implementation

The optimization problem for the $\sim_G$-PID, using projected gradient descent with RProp (mentioned in Section 3), is implemented as follows:

1. Let $\Sigma := \Sigma_{X,Y|M}$ be shorthand for the optimization variable, and let $\mathrm{Proj}(\cdot)$ represent the projection operator defined in Prop. 2. Let $\Sigma^{(i)}$ represent the value of $\Sigma$ at iteration $i$ of the optimization. Initialize $\Sigma^{(0)} = \mathrm{Proj}(H_X H_Y^+)$, where $H_Y^+$ is the pseudoinverse of $H_Y$.

2. Evaluate the objective and the gradient as defined in Prop. 2, at the current value of $\Sigma^{(i)}$. Compute the sign of (each element of) the gradient,

$$\psi(\Sigma^{(i)}) := \mathrm{Sgn}\big(\nabla f(\Sigma^{(i)})\big). \tag{59}$$

When computing the objective and the gradient, add a small regularization term to the computation of $S^{-1}$ (as defined in Prop. 2): $S^{-1} = \big((1+\epsilon)I - \Sigma\Sigma^{\mathsf{T}}\big)^{-1}$, where we take $\epsilon = 10^{-7}$.

3. Update:

$$\Sigma^{(i+1)} = \mathrm{Proj}\big(\Sigma^{(i)} - \alpha^i \eta^{(i)} \odot \psi(\Sigma^{(i)})\big), \tag{60}$$

where $\mathrm{Proj}$ is the projection operator defined in Prop. 2; $\eta^{(i)}$ is a time-varying learning rate vector of the same dimension as $\Sigma$, describing the learning rate for each element of $\Sigma$; $\odot$ represents an element-wise (or Hadamard) product between vectors; and $\alpha := 0.999$ is a constant, which when raised to the power of $i$, imposes a slow overall decay of the learning rate to promote convergence.

The matrix inverses embedded in the projection operator are also regularized by modifying Equation (58) as follows:

$$\Sigma_{X,Y|M}^{proj} = \big(\gamma I + \overline{\Sigma}_{X|M}^{1/2}\big)^{-1} \overline{\Sigma}_{XY|M} \big(\gamma I + \overline{\Sigma}_{Y|M}^{1/2}\big)^{-1}, \tag{61}$$

where $\gamma$ is slowly increased from $10^{-12}$, by a factor of 10 in each step, until $g(\Sigma_{X,Y|M}^{proj}) \succcurlyeq 0$.

4. $\eta^{(0)}$ is initialized to $10^{-3}$ and $\eta^{(i)}$ is updated as follows:

$$\eta^{(i+1)} = \eta^{(i)} \odot \beta^{-\psi(\Sigma^{(i+1)}) \odot \psi(\Sigma^{(i)})}, \tag{62}$$

where $\beta := 0.9$ is a constant that determines how fast the learning rate increases or decreases; and all operations are carried out element-wise. Note that when some element of the gradient changes in sign, that element of $-\psi(\Sigma^{(i+1)}) \odot \psi(\Sigma^{(i)})$ will be positive, resulting in a decrease in that element of $\eta^{(i)}$. On the other hand, if the sign of some element of the gradient remains the same, then the learning rate for that component will increase by a factor of $1/0.9$.

5. Stop when the absolute differences between the current objective and the previous objectives from the last 20 consecutive iterations are all less than $10^{-6}$ ("patience"), or when the maximum number of iterations is exceeded (set to $10^4$ iterations).

## A.5  Computational Complexity of the $\sim_G$-PID

Suppose for simplicity that $M$, $X$ and $Y$ all have the same dimensionality $d \coloneqq d_M = d_X = d_Y$. Then, $\Sigma_{MXY}$ and all its submatrices have a side of dimensionality $\mathcal{O}(d)$. Thus, the computational complexity of each gradient descent iteration is determined by the complexity of the matrix operations in equations (10)–(13). These operations include matrix multiplication, matrix inverses, $\log \det(\cdot)$ and eigenvalue decomposition, which have a worst case complexity of $\mathcal{O}(d^3)$.

Matrix multiplication can potentially be performed at lower complexity for large matrix sizes (e.g., see Strassen [42]), which has downstream implications for each of the other operations as well. However, for our purposes, we take the computational complexity of the objective, the gradient and the projection operator to all be $\mathcal{O}(d^3)$.

# B  Supplementary Material for Section 4

First, observe that by subtracting equation (2) from equation (1), we have

$$
\begin{aligned}
I(M;(X,Y)) - I(M;X) &= UI_Y + SI \\
\Rightarrow \qquad I(M;Y\,|\,X) &= UI_Y + SI.
\end{aligned}
\tag{63}
$$

Similarly, subtracting equations (1) and (3), we get that $I(M;X\,|\,Y) = UI_X + SI$. These two equations hold in general, and will be used in what follows.

## B.1  Details and Derivations for Examples in Section 4

**Example 2** (Pure uniqueness)**.**

$$
\begin{aligned}
M &\sim \mathcal{N}(0,1) & &\tag{64}\\
X &= M + N_X & N_X, N_Y &\sim \text{ i.i.d. } \mathcal{N}(0,1) \tag{65}\\
Y &= N_Y & (N_X, N_Y) &\perp\!\!\!\perp M \tag{66}
\end{aligned}
$$

*Derivation of PID values in Example 2.*

$$
\begin{aligned}
Y \perp\!\!\!\perp M \quad &\Rightarrow \quad I(M;Y) = 0 \tag{67}\\
&\Rightarrow \quad UI_Y + RI = 0 \tag{68}\\
UI_Y, RI \ge 0 \quad &\Rightarrow \quad UI_Y = RI = 0 \tag{69}\\
&\Rightarrow \quad UI_X = I(M;X) \tag{70}\\
&\Rightarrow \quad SI = I(M;(X,Y)) - I(M;X) \tag{71}\\
&\qquad\qquad = I(M;X) - I(M;X) = 0. \tag{72}
\end{aligned}
$$

$\square$

**Example 3** (Pure redundancy)**.**

$$
\begin{aligned}
M &\sim \mathcal{N}(0,1) & &\tag{73}\\
X &= M + N_X & N_X &\sim \mathcal{N}(0,1) \tag{74}\\
Y &= M + N_X & N_X &\perp\!\!\!\perp M \tag{75}
\end{aligned}
$$

*Derivation of PID values in Example 3.*

$$
\begin{aligned}
I(M;X\,|\,Y) = 0 \quad &\Rightarrow \quad UI_X + SI = 0 \tag{76}\\
I(M;Y\,|\,X) = 0 \quad &\Rightarrow \quad UI_Y + SI = 0 \tag{77}\\
&\Rightarrow \quad RI = I(M;(X,Y)) = I(M;X). \tag{78}
\end{aligned}
$$

$\square$

**Example 4** (Pure synergy)**.**

$$
\begin{aligned}
M &\sim \mathcal{N}(0,1) & &\tag{79}\\
X &= M + N_X & N_X &\sim \mathcal{N}(0,\sigma^2) \tag{80}
\end{aligned}
$$

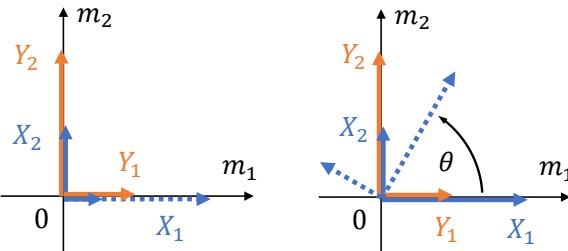

Figure 7: Diagrams explaining Examples 8 and 9. See App. B.2 for details.

$$Y = N_X \qquad\qquad N_X \perp\!\!\!\perp M \qquad\qquad (81)$$

*Derivation of PID values in Example 4.*

$$
\begin{aligned}
Y \perp\!\!\!\perp M &\Rightarrow & I(M;Y) &= 0 & (82)\\
&\Rightarrow & UI_Y + RI &= 0 & (83)\\
UI_Y, RI \geq 0 &\Rightarrow & UI_Y = RI &= 0 & (84)\\
&\Rightarrow & UI_X &= I(M;X) & (85)\\
&\Rightarrow & SI &= I(M;(X,Y)) - I(M;X) & (86)\\
& & &= \infty - I(M;X) = \infty. & (87)
\end{aligned}
$$

$\square$

It should be noted that certain nuances have been omitted in discussing Examples 2–4 above. For instance, in Example 3, $\Sigma_{XY|M}$ is rank deficient and hence non-invertible, which would be an issue when computing the objective in Equation (9). Also, in Example 4, $I(M;(X,Y)) = \infty$, however this could be corrected by adding some noise to either $X$ or $Y$ so that their difference is a noisy representation of $M$.

**Example 5** (Unique and redundant information).

$$
\begin{aligned}
M &\sim \mathcal{N}(0,1) & & & & & (88)\\
X &= M + N_X & N_X &\sim \mathcal{N}(0,1) & N_X &\perp\!\!\!\perp M & (89)\\
Y &= M + N_X + N_Y' & N_Y' &\sim \mathcal{N}(0,\sigma^2) & N_Y' &\perp\!\!\!\perp (N_X, M) & (90)
\end{aligned}
$$

*Derivation of PID values in Example 5.* Essentially, $X$ is a noisy representation of $M$, while $Y$ is a noisy representation of $X$. Since $M$—$X$—$Y$ forms a Markov chain, $I(M;Y\,|\,X) = 0$, and hence $UI_Y = SI = 0$. When $\sigma^2 = 0$, this example reduces to Example 3 with only redundancy being present. For any finite non-zero value of $\sigma^2$, both $RI$ and $UI_X$ are present and are non-zero. Since $M$ is scalar, the redundancy for the $\sim$-PID is identical to the MMI-PID's redundancy [13]:

$$RI = \min\{I(M;X), I(M;Y)\} = I(M;Y), \qquad (91)$$

since $I(M;Y) < I(M;X)$, by the data processing inequality. At the limit when $\sigma^2 \to \infty$, $I(M;Y) \to 0$, and therefore $RI \to 0$, while $UI_X$ will become equal to $I(M;X)$. $\square$

**Example 6** (Unique and synergistic information).

$$
\begin{aligned}
M &\sim \mathcal{N}(0,1) & & & & & (92)\\
X &= M + N_X & N_X, N_Y &\sim \mathcal{N}(0,\sigma^2), & (N_X, N_Y) &\perp\!\!\!\perp M & (93)\\
Y &= N_Y & \mathrm{Corr}(N_X, N_Y) &= \rho & & & (94)
\end{aligned}
$$

*Derivation of PID values in Example 6.* When $\rho = 1$ and $\sigma^2 \to \infty$, this example reduces to Example 4, with only synergy being present. In general, $Y \perp\!\!\!\perp M$, therefore, $I(M;Y) = UI_Y + RI = 0$, meaning $UI_Y = RI = 0$. For any finite value of $\sigma^2$, $X$ will have some unique information about $M$ given by $UI_X = I(M;X) > 0$. Correspondingly, $SI = I(M;X\,|\,Y) - UI_X =$

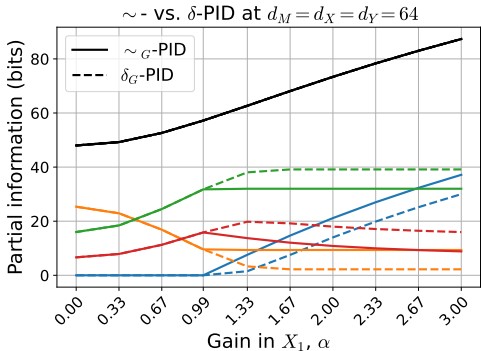

Figure 8: A comparison of the $\sim_G$- and $\delta_G$-PIDs at a dimensionality of $d = 64$. The plot is virtually identical to that in Fig. 2 (which uses $d = 2$), except that the $y$-axis is scaled up by a factor of 32.

$I(M; X \,|\, Y) - I(M; X)$. When $\rho = 0$, $I(M; X \,|\, Y) = I(M; X)$ and therefore $SI = 0$. As $\rho \to 1$, $X - Y \to M$; so the total mutual information $I(M; (X, Y)) \to \infty$, driven by synergy growing unbounded, while the unique component remains finite at $I(M; X)$. □

**Example 7** (Redundant and synergistic information)**.**

$$M \sim \mathcal{N}(0, 1) \tag{95}$$
$$X = M + N_X \qquad\qquad N_X, N_Y \sim \mathcal{N}(0, 1), \qquad (N_X, N_Y) \perp\!\!\!\perp M \tag{96}$$
$$Y = M + N_Y \qquad \mathrm{Corr}(N_X, N_Y) = \rho \tag{97}$$

*Derivation of PID values in Example 7.* When $\rho = 1$, we once again reduce to Example 3 with only redundancy. When $\rho < 1$, we cannot infer the PID values using Equation (1) and non-negativity alone, since none of the individual mutual information values (or conditional mutual information values) go to zero. Instead, we can determine the redundancy using the MMI-PID since $M$ is scalar. Note that $I(M; X)$ and $I(M; Y)$ are both equal by symmetry, and thus equal to $RI$. This also implies that both $UI_X$ and $UI_Y$ must be equal to zero. As $\rho$ reduces, the two channels $X$ and $Y$ have noisy representations of $M$ with increasingly independent noise terms. Therefore, their average, $(X + Y)/2$ will be more informative about $M$ than either one of them individually, meaning that $X$ and $Y$ jointly contain more information than any one individually. This extra information about $M$ is synergistic, given by $SI = I(M; X \,|\, Y)$, and increases as $\rho$ decreases, attaining its maximum possible value at $\rho = 0$. □

## B.2   Diagrams Explaining Examples 8 and 9

Examples 8 and 9 can be understood diagrammatically as shown in Fig. 7(l) and Fig. 7(r), respectively. In both diagrams, we represent the two-dimensional plane describing $M$, with axes $m_1$ and $m_2$. The colored vectors shown on this plane represent $H_X$ and $H_Y$, i.e., the gain with which $X$ and $Y$ represent each value of $M$. For example, $Y_2$ captures only $M_2$, with a gain corresponding to its length. The gains are directly representative of the signal-to-noise ratio (and hence the amount of information) in each variable, since the noise in each variable is i.i.d., with unit variance. In Example 8, the gain in $X_1$ is variable, while in Example 9, the angle at which $X_1$ and $X_2$ sample $M_1$ and $M_2$ is variable.

## B.3   Comparison of the $\sim_G$- and $\delta_G$-PIDs in Example 8 at Higher Dimensionality

For completeness, we also present a comparison between the $\sim_G$- and $\delta_G$-PIDs for Example 8 at a higher dimensionality of $d = 64$. The results can be found in Fig. 8, and show that both PIDs scale proportionally. We use a gain of $\alpha = 0.99$ in place of $\alpha = 1$, since a gain of unity causes certain matrices to become ill-conditioned.

Although the $\delta_G$-PID does not obey the additivity property in general (which is why it does not agree with the $\sim_G$-PID at gains of $\alpha > 1$), it appears to double along with the $\sim_G$-PID (i.e., additivity across identical independent copies appears to hold, at least in this specific example).

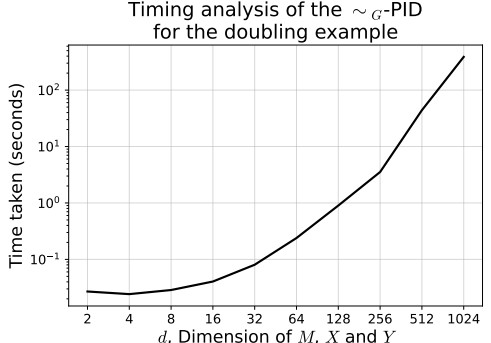

Figure 9: A plot showing the time taken to compute the $\sim_G$-PID at different dimensionalities in Example 10. This plot extends what is shown in the right-most plot of Fig. 3 for the $\sim_G$-PID.

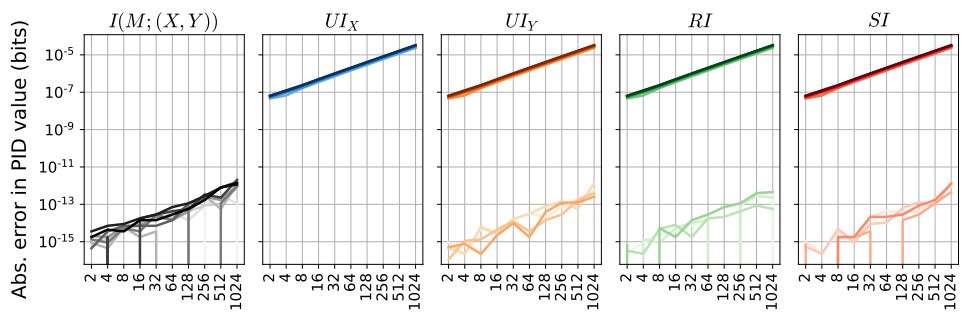

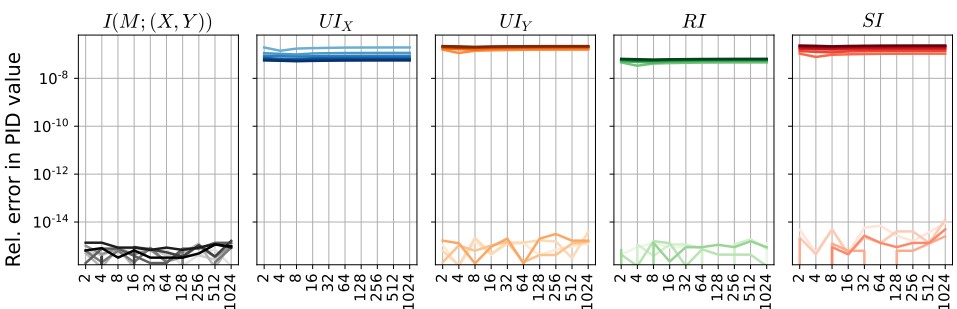

Figure 10: Absolute (top) and relative (bottom) errors in computed PID values from Example 10.

### B.4 Run Time Analysis of the $\sim_G$-PID in Example 10

In Fig. 9, we present a complete picture of the time taken to compute the $\sim_G$-PID, as a function of dimensionality $d$, in Example 10. Note that we cannot compare with the $\delta_G$-PID beyond a dimensionality of $d = 64$, because the computation for the $\delta_G$-PID failed in our setup for $d > 64$. Neither the $\delta_G$-PID nor the $\sim_G$-PID were carefully profiled and optimized, therefore, the runtime analysis in both Fig. 9 and Fig. 3 should be considered preliminary. We leave a more thorough comparison of the run time of both these methods to future work.

### B.5 Absolute and Relative Errors in Example 10

Figure 10 shows how the absolute and relative errors in PID values scale with increasing dimensionality in Example 10. The absolute errors increase in proportion to dimensionality, starting under $10^{-7}$

at $d = 2$ and remaining under $10^{-4}$ at $d = 1024$. The relative errors are all roughly constant, and remain under $10^{-6}$.

# C   Supplementary Material for Section 5

## C.1   Implementation Details for Bias-correction

We use a number of different setups based on sampling from random connectivity matrices for bias correction in Section 5. All of these setups assume that $d_X = d_Y$.

The **both-unique**, **fully-redundant** and **high-synergy** setups have the following in common:

$$\Sigma_M = I \tag{98}$$
$$\Sigma_{X|M} = I \tag{99}$$
$$\Sigma_{Y|M} = I \tag{100}$$
$$\Sigma_{MXY} = \begin{bmatrix} I & H_X^\mathsf{T} & H_Y^\mathsf{T} \\ H_X & H_X H_X^\mathsf{T} + \Sigma_{X|M} & H_X H_Y^\mathsf{T} + \Sigma_W \\ H_X & H_Y H_X^\mathsf{T} + \Sigma_W^\mathsf{T} & H_Y H_Y^\mathsf{T} + \Sigma_{Y|M} \end{bmatrix}. \tag{101}$$

Also, the elements of $H_X$ are either zero or one, $H_X(i,j) \sim$ i.i.d. Ber(0.1). These three setups differ in their definitions of $H_Y$ (the channel gain from $M$ to $Y$) and $\Sigma_W$ (which controls the extent of correlation between $X$ and $Y$).

The **both-unique** setup draws $H_Y(i,j) \sim$ i.i.d. Ber(0.1), with all elements of $H_Y$ independent of the elements of $H_X$, and sets $\Sigma_W = 0$.

The **fully-redundant** setup is similar to Example 7, by setting $H_Y = H_X$ and $\Sigma_W = 0.9I$ (note that $\Sigma_W$ is square, since $d_X = d_Y$). By keeping $\Sigma_W$ close to the identity matrix, we are effectively in the regime with high correlation $\rho$ in Example 7. This allows us to come close to emulating Example 3, without suffering from the issue of non-invertibility of $\Sigma_{XY|M}$, mentioned in App. B.

The **high-synergy** setup is similar to Example 6, by setting $H_Y = 0$ and $\Sigma_W = 0.8I$. As with the fully-redundant setup, by keeping $\Sigma_W$ close to the identity matrix, we are in the high-$\rho$ regime. This allows us to come close to emulating Example 4, while not making the synergy or the total mutual information infinite.

The **zero-synergy** setup is similar to Example 5, and uses the following setup:

$$\Sigma_M = \Sigma_{X|M} = I \tag{102}$$
$$\Sigma_X = H_X H_X^\mathsf{T} + \Sigma_{X|M} \tag{103}$$
$$\Sigma_{Y|X} = I \tag{104}$$
$$\Sigma_{MXY} = \begin{bmatrix} I & H_X^\mathsf{T} & H_Y^\mathsf{T} \\ H_X & \Sigma_X & \Sigma_X H_Y'^\mathsf{T} \\ H_X & H_Y' \Sigma_X^\mathsf{T} & H_Y' \Sigma_X H_Y'^\mathsf{T} + \Sigma_{Y|X} \end{bmatrix}. \tag{105}$$

Here, $H_X(i,j) \sim$ i.i.d. Ber(0.1), while $H_Y = H_Y' H_X$, with $H_Y'(i,j) \sim$ i.i.d. Ber(0.1), $H_Y' \perp\!\!\!\perp H_X$. Defined this way, $M$—$X$—$Y$ form a Markov chain, ensuring that $I(M; X \mid Y) = 0$, so that $SI = 0$ (Refer equation (63)).

The **bit-of-all** setup is a combination of equal parts of the high-synergy and zero-synergy setups. The variables $X$ and $Y$ are swapped in the zero-synergy setup, so that both $X$ and $Y$ can have some unique information.

**Remark 3** (Rectification). In practice, we observed that the bias correction procedure prescribed in Definition 4 could lead to negative values for certain PID quantities. This occurred because the bias-corrected union information was not guaranteed to satisfy certain bounds, which we enforce below. To prevent the occurrence of negative PID values after bias-correction, we require a form of rectification:

$$\widetilde{I_G^\cup}\Big|_{\text{rect}}^{(1)} := \max\Big\{ \widetilde{I_G^\cup}\big|_{\text{bias-corr}} , \; \hat{I}(M; X)\big|_{\text{bias-corr}} , \; \hat{I}(M; Y)\big|_{\text{bias-corr}} \Big\} \tag{106}$$

$$\left.\widetilde{I_G^{\cup}}\right|_{\text{rect}}^{(2)} := \min\left\{\left.\widetilde{I_G^{\cup}}\right|_{\text{rect}}^{(1)}, \; \left.\hat{I}(M;X)\right|_{\text{bias-corr}} + \left.\hat{I}(M;Y)\right|_{\text{bias-corr}}, \; \left.\hat{I}(M;(X,Y))\right|_{\text{bias-corr}}\right\}, \quad (107)$$

where $\hat{I}(\cdot)|_{\text{bias-corr}}$ represents a bias-corrected mutual information estimate. After the second rectification equation above, the union information is bounded from below by the individual (bias-corrected) mutual information values, and bounded from above by the sum of the individual mutual information values, and by the total mutual information. $\qquad\square$

### C.2 Bias-correction Performance in Additional Setups and at Higher Dimensionality

Plots showing bias correction performance for all setups described in App. C.1 are shown in Fig. 11 for 10-dimensional, and in Fig. 12 for 20-dimensional $M$, $X$ and $Y$.

Of the setups we examine, only the case with both $X$ and $Y$ having purely unique information appears to have somewhat poor performance, where our bias-correction method appears to over-correct the bias in unique information, while insufficiently correcting the bias in redundancy and synergy.

### C.3 A Preliminary Analysis of the Variance of PID Estimates

In Figures 13 and 14, we present a preliminary analysis of the variance of our PID estimates using bootstrap. The figures represent the true distribution of the PID estimates over multiple sample draws, or over multiple bootstrap sample draws, in the form of box plots. In what follows, we colloquially refer to these box plots as "confidence intervals". The true "confidence intervals" were estimated using 100 runs of bias-corrected PID estimates, i.e., by drawing 100 different samples, each of size $n$. The bootstrap "confidence intervals" were estimated using 100 bootstrap samples that were *resampled* from a *single* randomly drawn sample of size $n$.

When correcting for bias in the PID estimates on bootstrap samples, we use the number of *unique* data points in each bootstrap sample in place of $n$ (refer Corollary 4), rather than the total sample size. This leads to more stable bootstrap-PID estimates.

The quality of the bootstrap "confidence interval" is affected greatly by the quality of the individual sample used for bootstrap resampling. Nevertheless, we observe a reasonable degree of qualitative agreement between the true "confidence interval" and the bootstrap "confidence interval", particularly as the sample size increases. Future work will assess confidence intervals with greater care, using well-defined metrics, and assess how well these confidence intervals are calibrated.

## D  Supplementary Material for Section 6

### D.1 Details Regarding the Multivariate Poisson Spike-count Simulation

We follow our previous paper [12], where this analysis was first presented. In this simulation, $M$ is two-dimensional, consisting of two independent and identically distributed Poisson random variables, $M_1$ and $M_2$. $X$ and $Y$ are each generated through a linear combination of Binomially thinning $M_1$ and $M_2$, along with some Poisson noise:

$$M_1, M_2 \sim \text{Poiss}(2) \tag{108}$$
$$X \sim \text{Binom}(M_1, \alpha) + \text{Binom}(M_2, 0.5) + \text{Poiss}(1) \tag{109}$$
$$Y \sim \text{Binom}(M_1, 0.5) + \text{Binom}(M_2, 0.5) + \text{Poiss}(1) \tag{110}$$

While our results in Fig. 5 show that our $\sim_G$-PID estimator comes closest to obtaining the same values as the estimator of Banerjee et al. [20], several remarks are warranted:

1. The method of Banerjee et al. [20] estimates the $\sim$-PID and not the $\delta$-PID. Since there is no discrete estimator for the $\delta$-PID, it is entirely possible that the $\delta_G$-PID is equally (or more) accurate with respect to its own true value. That is, the difference between the $\delta_G$-PID and the Banerjee et al. [20] estimator is entirely due to a difference in definitions;

2. We stated in the introduction that the $\delta_G$-PID estimates an *approximate* upper bound, whereas our $\sim_G$-PID estimates an *exact* upper bound. The difference due to the approximate nature of the $\delta_G$-PID upper bound is also not captured here, nor in Examples 8 and 9. In all these examples, we cannot tell how much of the difference is due to the difference in definitions and the approximate nature of the estimators.

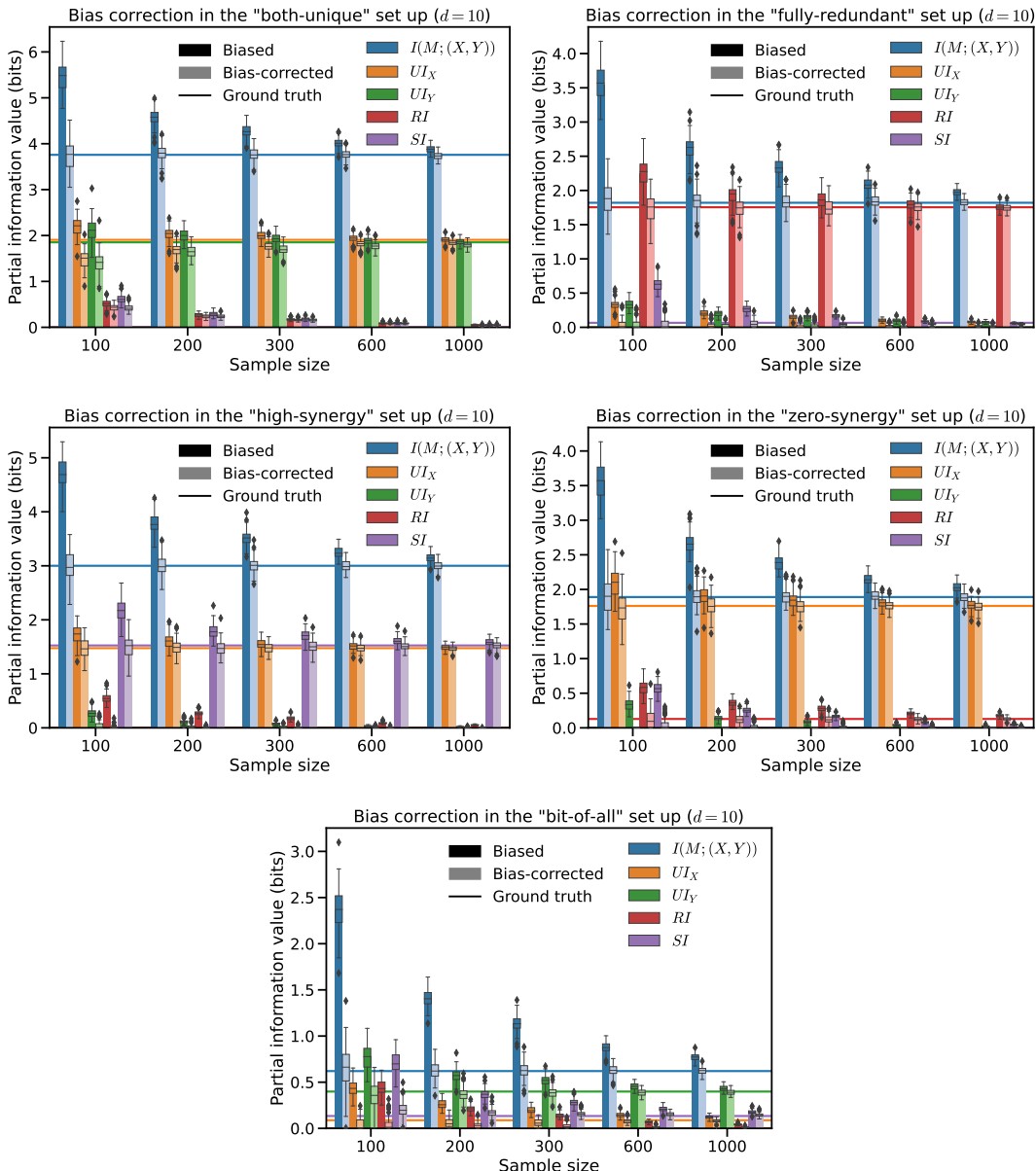

Figure 11: Bias correction for various setups described in App. C.1 with $d \coloneqq d_M = d_X = d_Y = 10$.

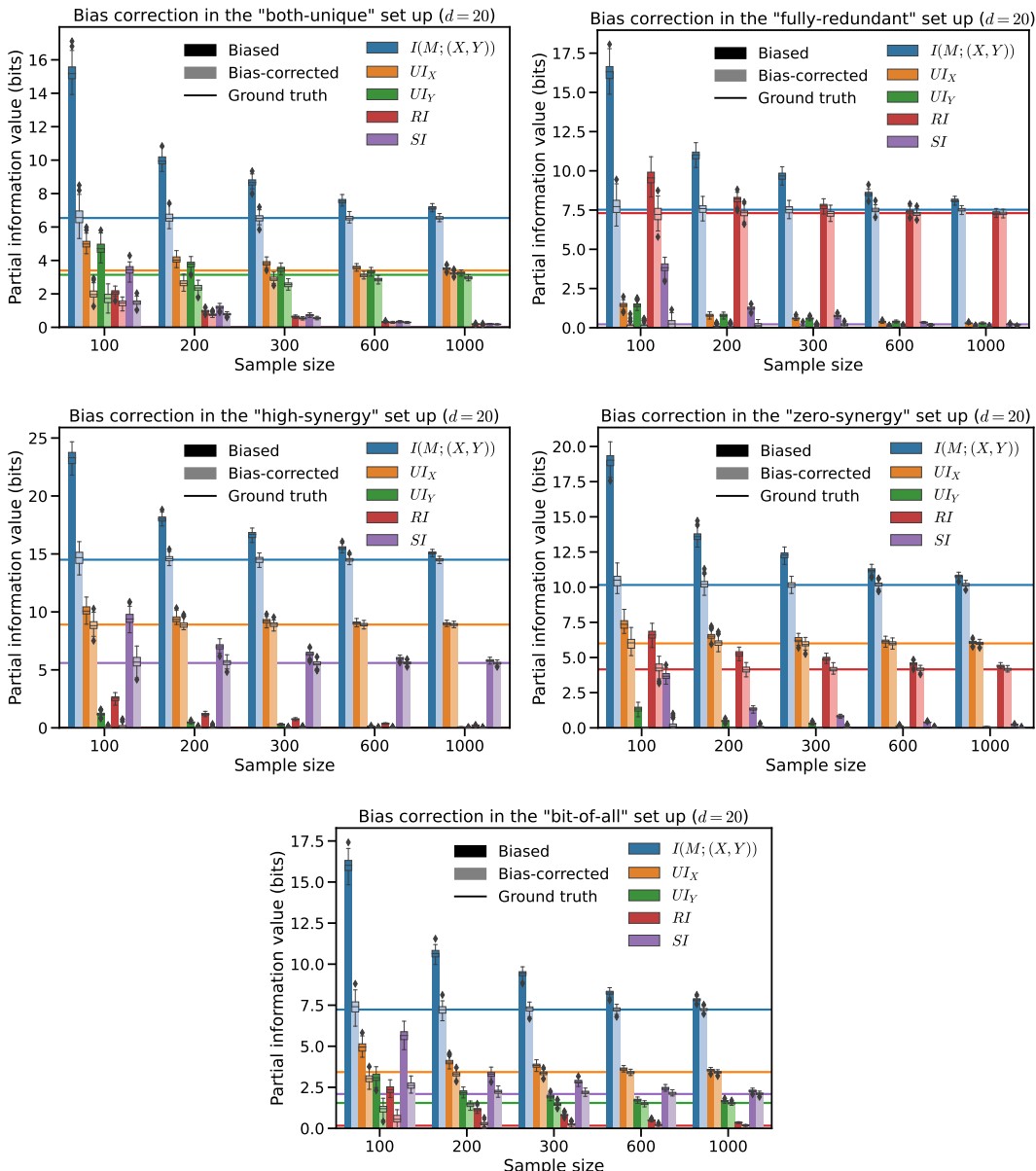

Figure 12: Bias correction for various setups described in App. C.1 with $d \coloneqq d_M = d_X = d_Y = 20$.

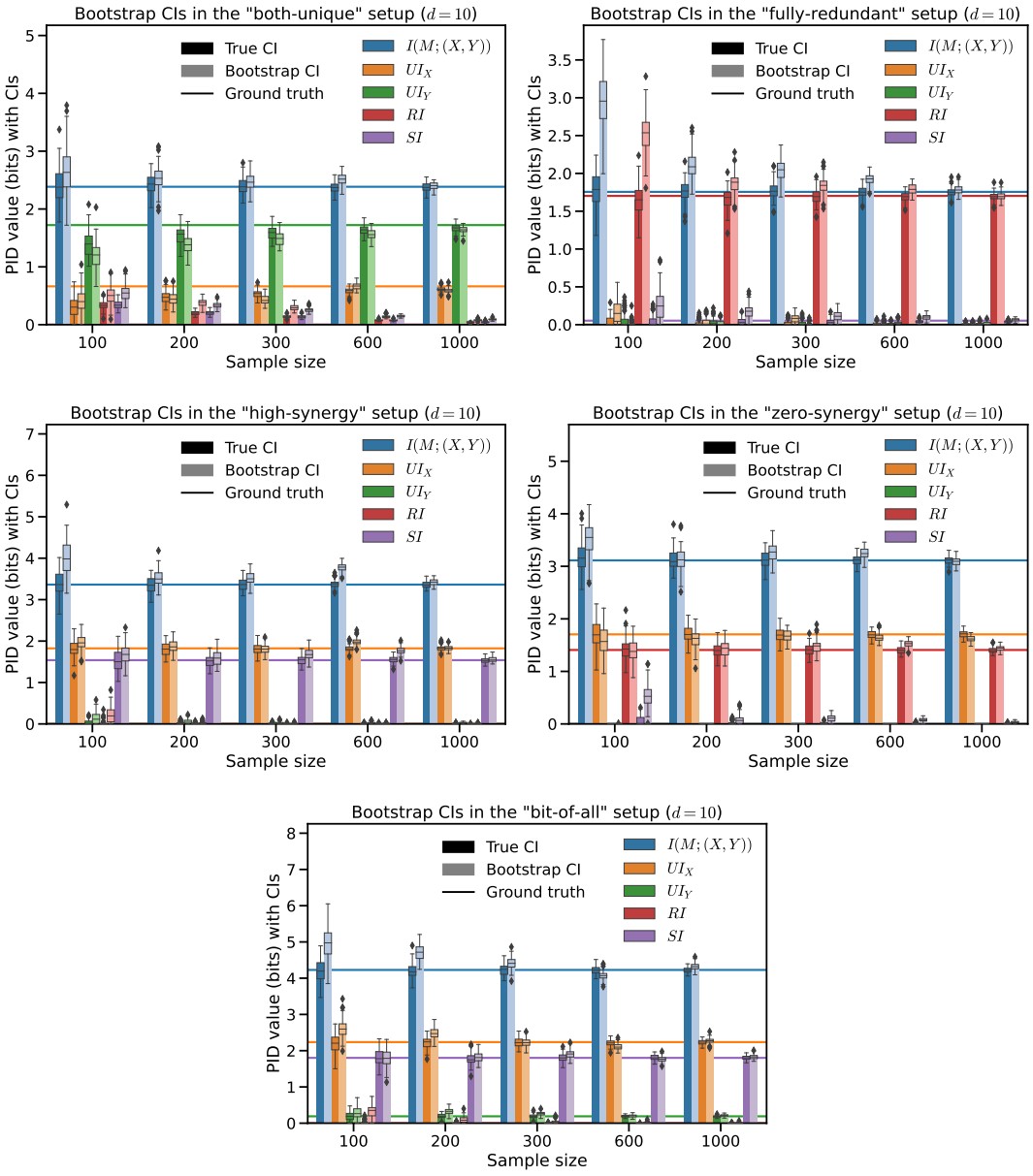

Figure 13: Bootstrap "confidence intervals" for various setups described in App. C.1 with $d := d_M = d_X = d_Y = 10$. Note that these are not true confidence intervals, but box-plot representations of the true variance of the estimator (over 100 runs) and the bootstrap estimate of the estimate's variance (from 100 bootstrap resamplings of a single random sample).

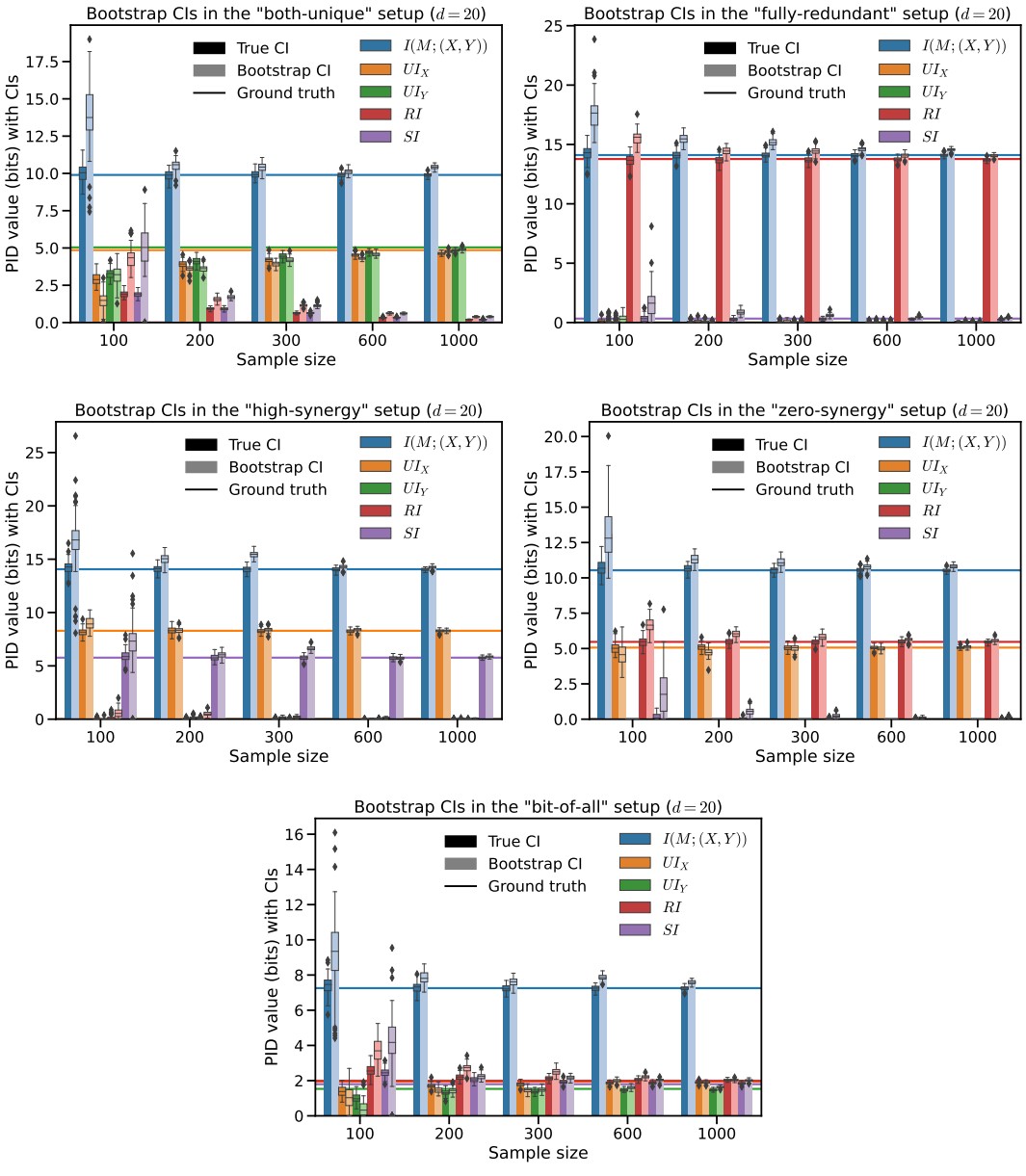

Figure 14: Bootstrap "confidence intervals" for various setups described in App. C.1 with $d := d_M = d_X = d_Y = 20$. Note that these are not true confidence intervals, but box-plot representations of the true variance of the estimator (over 100 runs) and the bootstrap estimate of the estimate's variance (from 100 bootstrap resamplings of a single random sample).

3. Finally, we do not know how accurate the "ground truth" itself is, since this is also assessed using the estimator of Banerjee et al. [20]. Differences between the various estimators could also be due to a poor estimate produced by the discrete PID estimator.

## D.2 Additional Simulations on Non-Gaussian Data

In order to evaluate how the $\sim_G$-PID performs on a greater variety of non-Gaussian distributions, we considered three more setups similar to the one described in App. D.1.

The first additional setup is a multivariate Binomial distribution:

$$M_1, M_2 \sim \text{Binom}(4, 0.5) \tag{111}$$
$$X \sim \text{Binom}(M_1, \alpha) + \text{Binom}(M_2, 0.5) + \text{Binom}(2, 0.5) \tag{112}$$
$$Y \sim \text{Binom}(M_1, 0.5) + \text{Binom}(M_2, 0.5) + \text{Binom}(2, 0.5) \tag{113}$$

This setup has the same mean as the multivariate Poisson distribution, but has a smaller variance and is therefore more peaked, further from zero. The multivariate Binomial distribution is closer to Gaussian than the multivariate Poisson, and indeed, we see good agreement between the $\sim_G$-PID and the "ground truth", as assessed by the estimator of Banerjee et al. [20] (see Fig. 15, top panels).

The next two setups are "zero-inflated" versions of the multivariate Poisson and Binomial distributions, which are produced by passing each of the aforementioned variables through a "Z-channel", with zero-out probability 0.3. That is,

$$M_1' = M_1 \times Z_{M_1} \qquad\qquad X' = X \times Z_X \tag{114}$$
$$M_2' = M_2 \times Z_{M_2} \qquad\qquad Y' = Y \times Z_Y \tag{115}$$

where $Z_i \sim$ i.i.d. $\text{Ber}(0.7)$, $i \in \{M_1, M_2, X, Y\}$. The primed versions of these variables have an additional probability mass at zero, with a probability of 0.3. This is said to better reflect the statistics of Calcium imaging [45], although we use a different base distribution than the one in the cited paper. The zero-inflated versions of the Poisson and Binomial distributions are bimodal, with one of the peaks at zero, and the Binomial version having more well separated peaks. For both zero-inflated versions, the $\sim_G$-PID performs poorly at recovering the absolute $\sim$-PID values (Fig. 15, middle-left and bottom-left panels). However, the relative PID values (normalized by the total mutual information) are closer to their true values (Fig. 15, middle-right and bottom-right panels), providing hope that a better mutual information estimate (e.g., [46, 47]) could correct the absolute PID values to some extent.

Further validation on non-Gaussian data is necessary to understand the impact of non-Gaussianity on $\sim_G$-PID estimates. However, a more extensive evaluation of non-Gaussian distributions (especially at higher dimensionalities) is made difficult by the unavailability of ground truth. The "ground truth" in these examples is obtained using the discrete $\sim$-PID estimator of Banerjee et al. [20] (which we assume is more accurate for discrete variables). Our method provides a better result than the $\delta$-PID and the MMI-PID in Fig. 5, at least part of which is probably due to the difference in definitions. Nevertheless, we believe these examples demonstrate that applying our method to non-Gaussian data need not be ruled out, although care should always be taken in interpreting results.

## D.3 Implementation Details of the Analysis Pipeline

The Visual Behavior Neuropixels data was analyzed as follows:

1. We selected mice that had at least 20 units in each brain region of interest. Only mice with both familiar and novel sessions were selected.

2. From each region we selected units of 'good' quality, with SNR at least 1, and with fewer than 1 inter-spike interval violations.

3. Trials were aligned to the start of each stimulus flash, and spikes were counted in bins of 50 ms, between 0 and 250 ms after stimulus onset (0-50 ms, 50-100 ms, etc.).

4. Trials corresponding to a non-change flash were defined as those that occurred between 4 and 10 flashes after the start of a behavioral trial, such that the image remained the same as the original image in this behavioral trial. Flashes corresponding to an omission, flashes after an omission, and flashes during which the animal licked, were all removed. Only flashes that occurred while

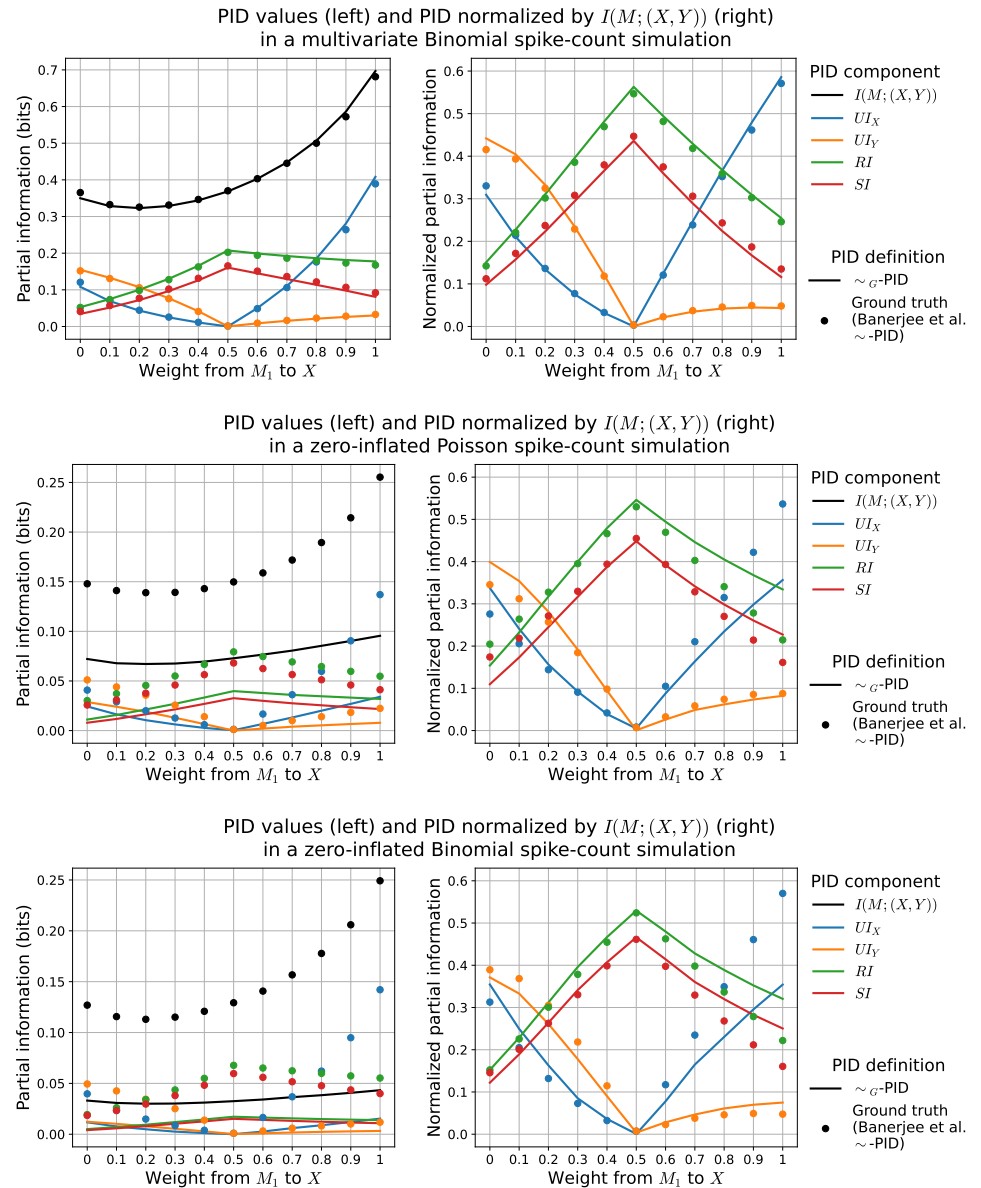

Figure 15: PID values and normalized PID values for three different non-Gaussian distributions, as described in App. D.2. The top, middle and bottom panels are respectively for a multivariate Binomial, a zero-inflated Poisson, and a zero-inflated Binomial distribution. The left column represents the PID values in bits, while the right column represents the PID values normalized by the total mutual information. All values are plotted as a function of the gain from $M_1$ to $X$ ($\alpha$) on the $x$-axis. An explanation of the results can be found in the text.

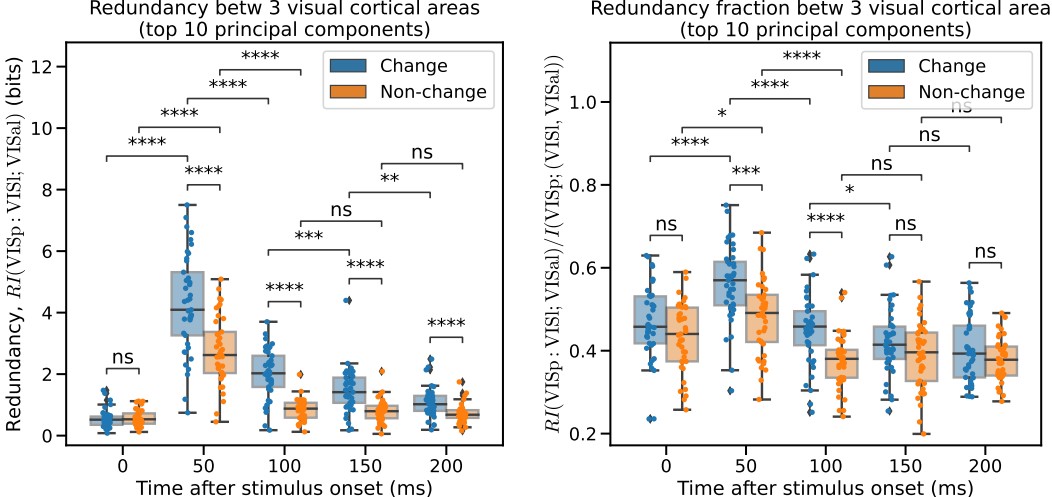

Figure 16: Redundancy about VISp activity between VISl and VISal, as a function of time, for flashes corresponding to an image change (blue) and flashes corresponding to a non-change (orange). Data points are across 41 mice. The plot on the left shows the raw redundancy in bits, while the plot on the right shows the redundancy normalized by the total mutual information.

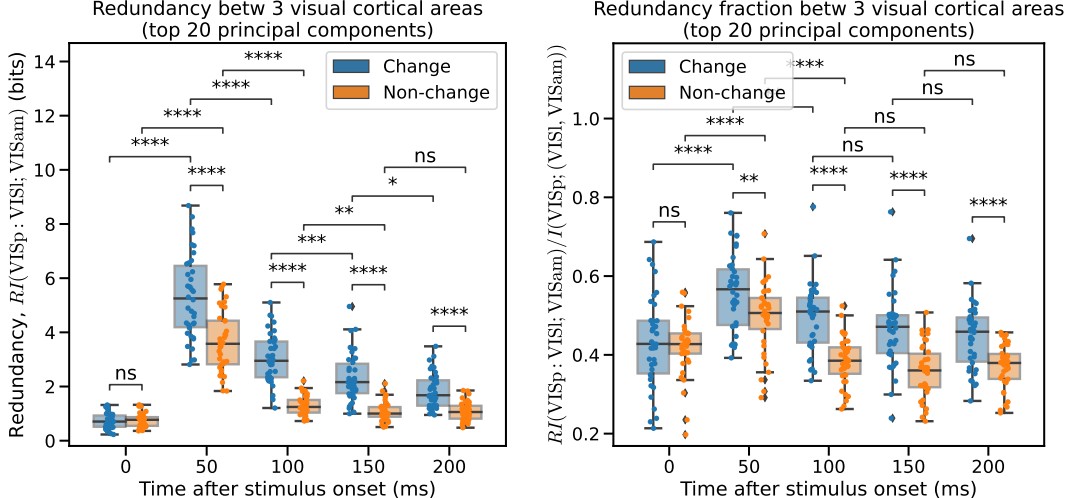

Figure 17: Redundancy about VISp activity between VISl and VISam, as a function of time, for flashes corresponding to an image change (blue) and flashes corresponding to a non-change (orange). Data points are across 38 mice. The plot on the left shows the raw redundancy in bits, while the plot on the right shows the redundancy normalized by the total mutual information.

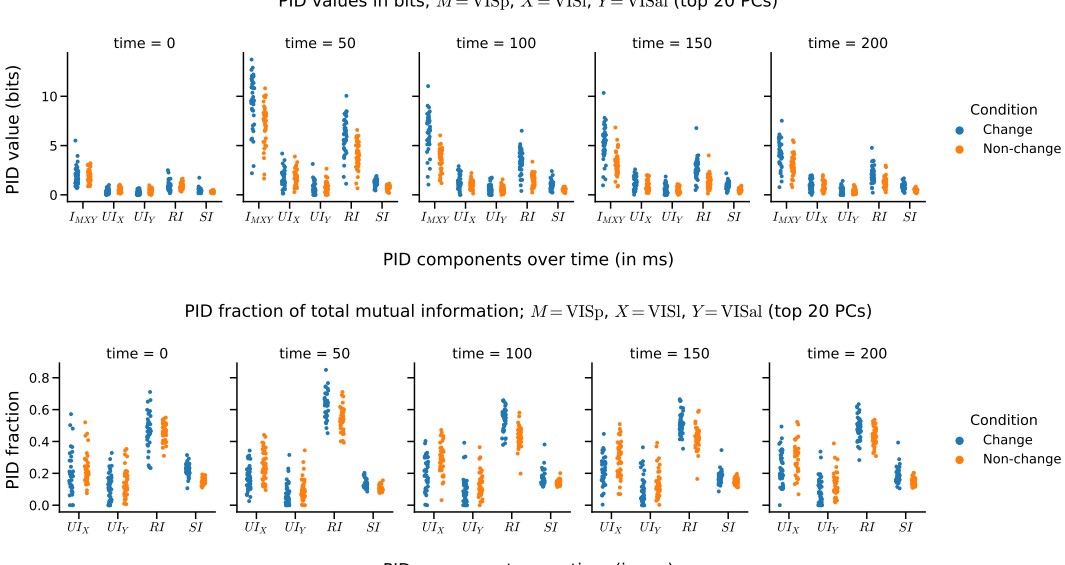

Figure 18: All PID components—about VISp activity, between VISl and VISal—in bits (top), and as a fraction of total mutual information (bottom), at various times after stimulus onset, for change and non-change flashes. Here, the label $I_{MXY}$ in the x-axis of the top plot refers to $I(M; (X, Y))$. These plots show that redundancy is the primary driver of mutual information.

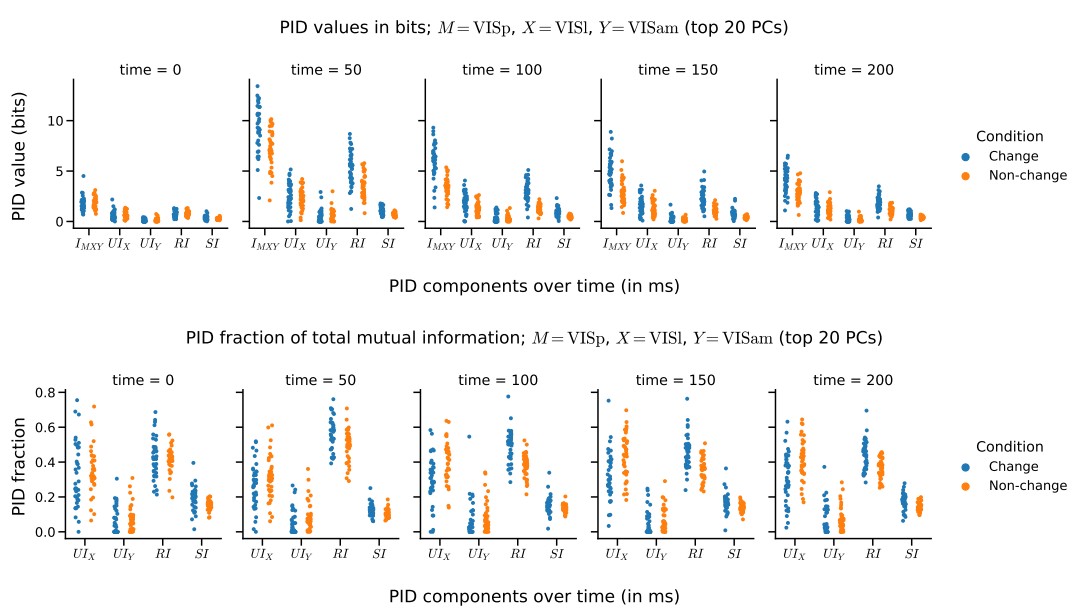

Figure 19: All PID components—about VISp activity, between VISl and VISam—in bits (top), and as a fraction of total mutual information (bottom), at various times after stimulus onset, for change and non-change flashes. Here, the label $I_{MXY}$ in the x-axis of the top plot refers to $I(M; (X, Y))$. These plots show that redundancy is the primary driver of mutual information.

the animal was engaged (as measured by an average reward rate of at least 2 rewards/min) were selected.

5. Trials corresponding to a change flash were defined as those during which an image change occurred, and the animal was engaged (as above).

6. The top 10 or 20 principal components of neural activity were selected at each time bin, and for each brain region under consideration. Principal component analysis was carried out using the Scikit-learn [48] package in Python.

7. $\sim_G$-PID estimates were computed on the covariance matrix between principal components across regions, for each time bin.

8. Data were aggregated across 40 or 41 mice for the figures with VISal, and across 38 mice for the figures with VISam.

9. Statistical significance was assessed using a two-sided unpaired Mann-Whitney-Wilcoxon test.

### D.4 Additional Results

Fig. 16 shows the redundancy about VISp activity, between VISl and VISal. It is identical to Fig. 6, except that it uses only the top-10 PCA components in each of the three visual cortical regions. This figure shows that the result presented in Fig. 6 gracefully degrades with a reduction in the number of PCA components. When using fewer PCA components, the increase in redundancy on change trials is still statistically significant, but the effect size is not as large, and not as sustained, as it is in Fig. 6.

Fig. 17 shows the redundancy about VISp activity between VISl and VISam, the latter of which is a different higher-order cortical region (see, e.g., [49]). This figure shows that the result in Fig. 6 can also be seen for another choice of higher-order cortical region (i.e., VISam rather than VISal).

Figures 18 and 19 show all PID components, not just the redundancy, for the same settings as in Figures 6 and 17 respectively. These show that redundancy is the dominant partial information component, and appears to be the main driver of changes in the overall mutual information. This justifies why we include only a plot of redundancy in Figs. 6, 16 and 17.

### D.5 The Necessity for PCA, and Maximizing Dimensionality

In our analysis of real neural data in Sec. 6, we used principal components analysis to reduce dimensionality, despite the fact that our method could scale to much larger dimensionalities. As mentioned briefly in Sec. 6, this was due to the limited number of trials we had. For many mice, there were a larger number of neurons in each region than trials with which to compute the covariance matrix. In other words, directly computing the covariance of the neural activity would have resulted in a covariance matrix that was rank-deficient. We used PCA to reduce dimensionality and ensure that we obtained a reasonable estimate of the covariance matrix, and to minimize the error in our PID estimate, which would naturally be higher with a greater number of PCA dimensions.

For completeness, we analyze here what would happen if we used the maximum possible number of PCA components for each mouse, while ensuring that the covariance matrix is not rank deficient. We also ensure that we had the same number of components across all regions and across both change and non-change conditions (within each mouse), so as to perform a fair comparison. This gives us on average 53±16 PCA components, with a minimum of 22 and a maximum of 84 PCA components. We find that the results of higher and more sustained redundancy on change flashes continues to hold when using the maximum possible number of PCA components for each mouse (see Fig. 20). However, as expected, there is much greater variance across mice, possibly due to variability in the number of PCA components chosen across mice, and due to greater errors in our PID estimates. For the main results presented in Fig. 6, in order to be consistent across mice, we rounded down from the minimum and used the top 20 PCA components for all mice.

### D.6 Differences between Change and Non-change Conditions are not an Artifact of Bias-correction

The number of trials corresponding to change flashes is much smaller than the number of trials corresponding to non-change flashes. Accordingly, the sample size used to estimate the covariance matrix is different in each of the two conditions. Bias correction was performed using the appropriate

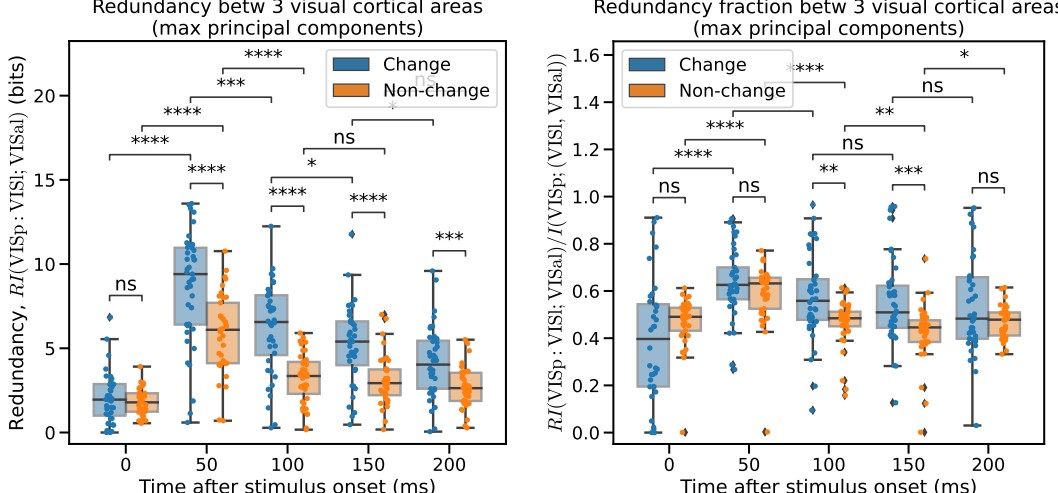

Figure 20: Redundancy about VISp activity between VISl and VISal, using the maximum possible number of PCA components for each mouse. Data points are across 42 mice, however, not all mice are represented at all points in time and for both conditions, since individual data points may not have converged due to ill-conditioned matrices. The plot on the left shows the raw redundancy in bits, while the plot on the right shows the redundancy normalized by the total mutual information. The increase in redundancy on change trials is still statistically significant, however, the errorbars have a greater spread for reasons described in the text.

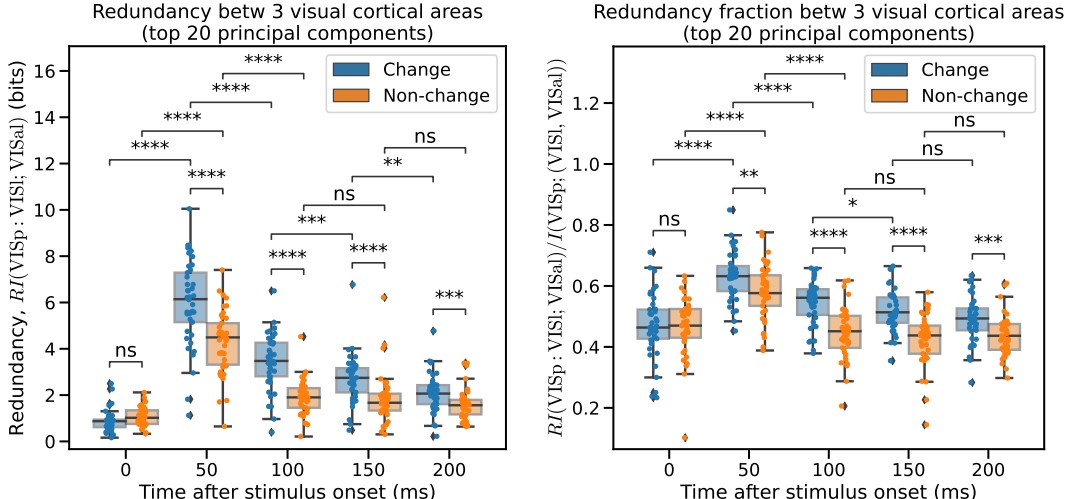

Figure 21: Redundancy about VISp activity between VISl and VISal, as a function of time, for flashes corresponding to an image change (blue) and flashes corresponding to a non-change (orange) with an equal number of samples. Data points are across 41 mice. The plot on the left shows the raw redundancy in bits, while the plot on the right shows the redundancy normalized by the total mutual information. The observations made in the other figures continue to hold; the differences seen between the two conditions are, therefore, not a result of differences in sample size.

sample size; however, as noted in Section 5, our bias correction process is not perfect, and may leave some residual bias.

In order to show that the results we observed were not an artifact of differences in residual bias caused by different sample sizes, we randomly subsampled the non-change flashes to produce a dataset with equal numbers of trials for change and non-change flashes. Repeating the analysis as before, we found that our conclusions continued to hold even in the setting where both conditions have equal sample sizes, as shown in Figure 21.

### D.7 A Note on Inhomogeneous Poisson Processes

Inhomogeneous Poisson processes are those whose mean firing- (or "emission-") rates change with time. Our method for estimating the PID does not consider the temporal characteristics of the original signal. Rather, the data analyst can choose the random variables $M$, $X$ and $Y$ to span some time range (or different time ranges) as they please.

In our analysis of real neural data in Sec. 6, we counted the number of spikes in a 50-125 ms window after stimulus onset. Even if the underlying spiking process was an inhomogeneous Poisson process, this spike count would be Poisson distributed, with a mean given by the integral of the emission rate over the fixed window. In general, while analyzing data and computing PID values, one will have to be aware of severe inhomogeneities (e.g., if the distribution changes not just in the same way in each trial, but changes across trials), and account for them separately. For example, in our analysis, we excluded time periods when the mice were not "engaged" in the task, as defined by not actively consuming rewards at a rate of at least 2 rewards per minute.

## E  Compute Configuration Used and Code Availability

All analyses were performed on a workstation equipped with an Intel Core i7-10700KF CPU with 8 cores (16 threads), 48 GiB of RAM and data stored on a 1 TB PCIe NVMe solid state drive.

Analysis of the Visual Behavior Neuropixels data (for 84 sessions) with $d = 20$ took approximately 9 minutes to run. This included loading data for each session and computing 840 PID values on $60 \times 60$ covariance matrices, implying an average run-time of about 0.64s for each PID estimate (including amortized data-load time).

All code used to compute and estimate the $\sim_G$-PID and correct for bias, including all examples in this paper and code for neural data analysis, is available on GitHub [36].

## Additional References

[43] Thomas M Cover and Joy A Thomas. *Elements of Information Theory*. John Wiley & Sons, 2012.

[44] K. B. Petersen and M. S. Pedersen. *The Matrix Cookbook*. Technical University of Denmark, 2012. URL http://www2.compute.dtu.dk/pubdb/pubs/3274-full.html. Version: November 15, 2012.

[45] Xue-Xin Wei, Ding Zhou, Andres Grosmark, Zaki Ajabi, Fraser Sparks, Pengcheng Zhou, Mark Brandon, Attila Losonczy, and Liam Paninski. A zero-inflated gamma model for post-deconvolved calcium imaging traces. *Neurons, Behavior, Data Analysis, and Theory*, 3(2):1–21, 2020.

[46] Alexander Kraskov, Harald Stögbauer, and Peter Grassberger. Estimating mutual information. *Physical review E*, 69(6):066138, 2004.

[47] Mohamed Ishmael Belghazi, Aristide Baratin, Sai Rajeshwar, Sherjil Ozair, Yoshua Bengio, Aaron Courville, and Devon Hjelm. Mutual information neural estimation. In *International conference on machine learning*, pages 531–540. PMLR, 2018.

[48] F. Pedregosa, G. Varoquaux, A. Gramfort, V. Michel, B. Thirion, O. Grisel, M. Blondel, P. Prettenhofer, R. Weiss, V. Dubourg, J. Vanderplas, A. Passos, D. Cournapeau, M. Brucher, M. Perrot, and E. Duchesnay. Scikit-learn: Machine learning in Python. *Journal of Machine Learning Research*, 12:2825–2830, 2011.

[49] Joshua H Siegle, Xiaoxuan Jia, Séverine Durand, Sam Gale, Corbett Bennett, Nile Graddis, Greggory Heller, Tamina K Ramirez, Hannah Choi, Jennifer A Luviano, et al. Survey of spiking in the mouse visual system reveals functional hierarchy. *Nature*, 592(7852):86–92, 2021.

