# OpenReview forum: "Gaussian Partial Information Decomposition: Bias Correction and Application to High-dimensional Data"
_NeurIPS.cc/2023/Conference — NeurIPS 2023 spotlight_

### Official Review · Reviewer_YrjF · 2023-07-04

**Soundness:** 3 good
**Presentation:** 4 excellent
**Contribution:** 3 good
**Rating:** 8
**Confidence:** 3

**Summary:**

The paper proposed a new method for partial information decomposition (PID) on multivariate Gaussian distributions. The issue of bias was discussed, and a correction method was provided. The method was tested on synthetic canonical examples and real data.

**Strengths:**

1. The introduction clearly lays out the problem.
2. The paper is extremely well written with notations clearly defined.
3. The method extends prior work on PID with new properties.
4. The method is rigorously tested on simulated data.
5. The method solves an important problem, namely “the extent to which one region’s activity uniquely explains that of another, while excluding information corresponding to spontaneous behaviors” as stimulus could make two regions seem correlated.

**Weaknesses:**

1. The paper does not provide enough real data to show effectiveness in real applications.
2. Testing on higher dimensional settings would be important.

**Questions:**

1. Would non-modeled confounding factors affect the results?

Summary after rebuttal
The added simulations for testing higher dimensional data further justify the high score I gave.

**Limitations:**

The paper does not have negative societal impacts.

---

> ### Author Rebuttal · Authors · 2023-08-10
>
> **Weaknesses**
>
> > 1. The paper does not provide enough real data to show effectiveness in real applications.
>
> We thank the reviewer for raising this point. We have tried to ameliorate this concern by demonstrating how the method works with a larger number of neurons (i.e., PCA components) in the same dataset (Fig. 8 and 9 in the attached PDF). We have also demonstrated our method on more simulated non-Gaussian examples, to understand the extent to which our method applies in non-Gaussian settings (Figs. 1-4 in the attached PDF). The focus of our paper was showing that our method works well on examples where the ground truth was known, as well as showing a proof of concept on real data. A more extensive evaluation of our method on real datasets is certainly necessary, but we feel it is beyond the scope of this paper.
>
> > 2. Testing on higher dimensional settings would be important.
>
> We have increased the dimensionality of the simulated high-dimensional example with known ground truth (Example 10, and we find that our method begins to depart from the ground truth at a dimensionality of 512 (please see the overall rebuttal, and Fig. 5 in the attached PDF)
>
> **Questions**
>
> > Would non-modeled confounding factors affect the results?
>
> Our Gaussian PID depends only on the joint covariance matrix between M, X and Y. Different models of external confounding that result in the same covariance matrix will not affect the PID values obtained. However, if one or more of these confounding variables are included in M, X or Y, and the PID is then re-computed, that may change the PID values completely compared to the case where the confounding variable is not included.
>
> Confounders are also important to consider when interpreting PID values. Our ~_G-PID method is based on the covariance matrix between M, X and Y, and is thus a correlational quantity. We cannot make causal claims based on observed PID values. Many of the same caveats that apply to interpreting correlations would also apply to interpreting observed PID values. A detailed analysis of the connections between different structural causal models (e.g., see Peters et al. 2017, “Elements of Causal Inference”) and their PID profiles has not been undertaken in the literature, to our knowledge, but could be the subject of future work (which is also better enabled by this paper).

---

> > ### Comment · Reviewer_YrjF · 2023-08-15
> > **Response to Rebuttal**
> >
> > The added simulations for testing higher dimensional data further justify the high score I gave.

---

### Official Review · Reviewer_DDWz · 2023-07-05

**Soundness:** 3 good
**Presentation:** 3 good
**Contribution:** 2 fair
**Rating:** 6
**Confidence:** 3

**Summary:**

Partial Information Decompositions (PIDs) play an essential role in neuroscience research. One constraint of the broader usability of PIDs is the computational difficulty of computing PIDs for high-dimensional neural recordings. To address this concern, the authors propose a method to compute and estimate a PID efficiently. More specifically, they restrict the optimization space of PIDs to jointly Gaussian, which reduces the number of optimization variables, allowing them to compute PIDs for much higher dimensionalities of neural data. Then, the authors show their method could be written out in closed form and solved by projected gradient descent, and they use nine examples with increasing complexity to show their method has the ability to recover ground truth and stability over increasing dimensionality. The authors also claim it's the first time to correct the bias and variance of estimates in a heuristic way. Finally, the authors evaluate the performance of their method on both synthetic and real neural data.

**Strengths:**

* Computational scalability of ~PID with a basic property called additivity.
* The authors' exposition of their method and experiments is clear.
* The first time to raise the issue of bias in PID estimates.

**Weaknesses:**

* No analysis is done to show the stability of $\delta$-PID over increasing dimensionality. According to the paper, there are two differences between $\delta$-PID and $\sim$-PID. The first one is the "additivity" property, which is clearly shown by Examples 8-9 in Section 4. The second one is that $\sim$-PID uses an exact upper bound, while $\delta$-PID uses an approximate upper bound.
However, no analysis is performed to show how the second difference will affect the performance. In other words, people may be curious about the stability of $\delta$-PID over increasing dimensionality if we use examples with no "additivity" property ($\delta$-PID and $\sim$-PID both agress with the ground truth).
Currently, it's hard to see the differences between $\delta$-PID and $\sim$-PID when applying to high dimensional neural data. I think this point may determine whether the proposed contribution is timely and impactful or a solid technical upgrade without practical consequences in neuroscience.

**Questions:**

* Does $\delta$-PID also agree with the ground truth even when distributions are not Gaussian?
* Could you please discuss how the "additivity" property would affect the analysis of communication among brain regions?

**Limitations:**

I have highlighted technical limitations and weaknesses above. I have nothing further to add here.

---

> ### Author Rebuttal · Authors · 2023-08-10
>
> **Weaknesses:**
>
> > No analysis is done to show the stability of delta-PID over increasing dimensionality. According to the paper, there are two differences between delta-PID and ~-PID. The first one is the "additivity" property, which is clearly shown by Examples 8-9 in Section 4. The second one is that ~-PID uses an exact upper bound, while delta-PID uses an approximate upper bound.
> >
> > However, no analysis is performed to show how the second difference will affect the performance. In other words, people may be curious about the stability of delta-PID over increasing dimensionality if we use examples with no "additivity" property (delta-PID and ~-PID both agress with the ground truth).
> >
> > Currently, it's hard to see the differences between delta-PID and ~-PID when applying to high dimensional neural data. I think this point may determine whether the proposed contribution is timely and impactful or a solid technical upgrade without practical consequences in neuroscience.
>
> We thank the reviewer for raising this point. We have added a comparison with the delta-PID for a larger dimensionality (please see Fig. 6,7 in the attached PDF).
>
> Just as the delta-PID fails at a dimensionality of 2, it also fails at 64 (i.e., the error due to additivity extends to higher dimensions). Furthermore, we found that the delta-PID method begins to take a very large amount of time to compute (> 2 hours) at $d_M = d_X = d_Y = 128$, and completely exceeds the memory capacity of a 48GiB workstation at a dimensionality of 256. We have included a timing analysis (please see Fig.7 in the attached PDF) to show the difference in speed of the two methods; our method runs over 1000 times faster at a dimensionality of 64. We will add this analysis to the paper.
>
> It is much harder to measure the difference between the delta and ~-PIDs due to the difference caused by the delta-PID recovering an approximate upper bound. The cases where the delta-PID agrees with the ~-PID are points where the solution is known in closed form due to a theorem from [12]. In other words, these are “easy” cases. All non-trivial cases are the ones that are constructed using the additivity property. The effective result of this analysis is that the degree to which the ~-PID outperforms the delta-PID due to the fact that the latter’s upper bound is approximate is still unknown. We will mention this point in the revised paper.
>
> However, we still believe our method will be impactful and have practical consequences for neuroscience, because it works accurately at much higher dimensions, runs much faster, and due to the importance of the additivity property (as described below).
>
> **Questions:**
>
> > 1. Does delta-PID also agree with the ground truth even when distributions are not Gaussian?
>
> To the extent that we consider Banerjee et al.’s (ISIT 2018) method to provide ground truth, we show in the paper in Fig. 5 that the delta-PID does not agree with the ground truth to the same degree as the ~-PID for a non-Gaussian distribution. However, it should be noted that Banerjee et al.’s method uses the ~-PID definition. Thus any difference between the delta_G-PID (dashed line) and Banerjee et al.’s method (“ground truth”) could be a result of the difference in the two definitions. Unless we have a more accurate way to compute the delta-PID for discrete distributions, we would not be able to say whether the difference was purely due to the difference in definitions, or due to a difference in accuracy. For additional clarity, we will add this discussion to the paper.
>
> > 2. Could you please discuss how the "additivity" property would affect the analysis of communication among brain regions?
>
> Additivity is an extremely fundamental property: effectively, it states that the PID values of an isolated system should not depend on the PID values of another isolated system. Without additivity, it is not possible to examine systems in isolation, since broadening your view to include a different isolated system could change the PID values of the first system.
>
> Hypothetically, if we take two separate individuals receiving completely independent stimuli, and examine the PID between the activity in brain regions M, X and Y in each of their brains, then the effective unique information that X1 and X2 have about M1 and M2 with respect to Y1 and Y2 should be equal to the sum of the unique information in each of the individuals taken separately.
>
> As another idea, suppose we are trying to examine visual information flow and auditory information flow in a multi-sensory integration task. We may want to understand the degree to which the activity in the sub-regions of the visual and auditory systems depend on each other. A reasonable null model of independence between the two systems would be that the PID values of the joint system will be equal to the sum of the PID values in the two individual systems. Then, measuring the actual degree to which the joint PID value is not equal to the sum will be a meaningful measure of dependence between the systems. This would only be possible with a PID definition that _guaranteed_ additivity of independent sub-systems. However, since the delta-PID does not satisfy the additivity property, we cannot guarantee that the aforementioned null would be the correct null model, and we would not be able to perform such an analysis. We will add this example to the supplementary material.

---

> > ### Author Response · Authors · 2023-08-18
> > **Follow up comment to Reviewer DDWz**
> >
> > We request Reviewer DDWz to let us know if our responses have addressed their concerns, and to kindly consider whether our paper is now worthy of a better score.
> >
> > We also request the reviewer to get back to us soon, since author responses will be closed after August 21st, 1pm EDT.
> >
> > In addition to the overall rebuttal and the reviewer-specific rebuttal, we would also like to draw the reviewer's attention to the analysis posted in an official comment at the top of this page, which extends our simulation in Example 10 to 1024 dimensions in each of M, X and Y. This further shows how our method is more capable and applicable than the $\delta$-PID of [12], which fails at a dimensionality of 256 in the same experiment.

---

> > > ### Comment · Reviewer_DDWz · 2023-08-19
> > >
> > > Thank for your additional experiments and explanation. I will raise my socre to 6 for addressing my questions and concerns.

---

### Official Review · Reviewer_wULj · 2023-07-07

**Soundness:** 3 good
**Presentation:** 3 good
**Contribution:** 3 good
**Rating:** 6
**Confidence:** 3

**Summary:**

The authors propose a new, efficient method for computing Partial Information Decompositions (PIDs) on multivariate Gaussians. They build their approach around the $\sim$-PID approach, as this allows them to preserve an additivity property (allowing PIDs to be computed on independent systems separately and then added later). They present a number of canonical examples for Gaussians, shaw that their Gaussian PID works even when distributions are non-Gaussian, and show an example of its use on real neural data from the Allen Institute. Finally, they address the issue of bias in PID estimates, propose a bias-correction method, and evaluate it empirically.

**Strengths:**

The paper is very clearly written and the topics well explained. The literature review seems comprehensive, and although of somewhat specific topical interest, the work seems original and useful.

**Weaknesses:**

Given one of the major motivating factors given in the introduction is the need for efficient estimators of PID that can accommodate higher dimensional neural data, I was disappointed that in the end the authors chose to apply their method to a PCA-reduced version of the Allen Institute neuropixel probe data... wasn't the point to be able to address higher dimensional problems? It's thus not completely clear to me that this method, as is, has delivered on the promise of providing a PID approach more applicable to data with thousands of neurons than other PID methods.

**Questions:**

Can the authors method run on thousands of neurons, consistent with the original motivation they provide?
What are the limitations on dimensionality more explicitly for this method?
What is the computational complexity?
Does the author's method require using PCA or having high firing rate neurons?
How does it deal with low firing rate neurons or highly inhomogenous Poisson variables?

**Limitations:**

The authors adequately address limitations, although it would be good to hear more related to the questions above.

---

> ### Author Rebuttal · Authors · 2023-08-10
>
> **Weaknesses**
>
> > Given one of the major motivating factors given in the introduction is the need for efficient estimators of PID that can accommodate higher dimensional neural data, I was disappointed that in the end the authors chose to apply their method to a PCA-reduced version of the Allen Institute neuropixel probe data... wasn't the point to be able to address higher dimensional problems? It's thus not completely clear to me that this method, as is, has delivered on the promise of providing a PID approach more applicable to data with thousands of neurons than other PID methods.
>
> We thank the reviewer for raising this concern and providing us with an opportunity to address it. We have now performed the PID analysis on the Allen Institute data again, using a larger number of PCA components (as described below; Figs. 8,9 in attached PDF). We were forced to use PCA because there were often a larger number of neurons in each region, than the number of trials available for computing the covariance matrix (in other words, if we had directly computed the covariance on the neural activity, the covariance matrix would have been rank-deficient). We wanted to ensure that we obtained a reasonable estimate of the covariance, and minimize the error in our PID estimate, which would naturally be higher for higher PCA dimensions.
>
> In the new analysis, for each mouse in the dataset, we used as many PCA components as possible, based on the number of neurons in each region and the number of trials available for computing the covariance matrix. We used the maximum number of PCA components subject to these constraints, while using the same number of components across all regions and across change and non-change conditions (within each mouse), so as to perform a fair comparison. This gave us on average $53 \pm 16$ PCA components, with a minimum of 22 and a maximum of 84 PCA components. We find that the results of higher and more sustained redundancy on change flashes continue to hold in the new analysis (Figs. 8,9 in the attached PDF). However, as expected, there is much greater variance across mice, possibly due to variability in the number of PCA components chosen across mice, and due to greater errors in our PID estimates. We will add these new results to the revised supplementary material.
>
> For consistency across mice, we used a common basis of 10 or 20 PCA components across all mice (and across all regions and conditions). Fig. 6 in the paper had 10 PCA components; Figs. 13 and 14 in the supplementary material of the paper had 20 PCA components. We will also include the rationale for selecting up to 20 PCA components in the revised version of the paper.
>
> Apart from this, to show that we can go to higher dimensions, we have also extended our high-dimensional example (Example 10) to a dimensionality of 256 (equivalent to a total of 768 neurons across regions). It should also be noted that other PID methods do not come close (please see the overall rebuttal for comparisons): the discrete PID estimators are limited to low single-digit dimensions, while other methods do not show ground truth validation at high dimensions.
>
> **Questions**
>
> > Can the authors method run on thousands of neurons, consistent with the original motivation they provide? What are the limitations on dimensionality more explicitly for this method? What is the computational complexity? Does the author's method require using PCA or having high firing rate neurons? How does it deal with low firing rate neurons or highly inhomogenous Poisson variables?
>
> 1. We have re-run our analysis on as many dimensions as possible in the Allen Institute dataset, as described above. We have also increased the dimensionality in our simulated data experiment and shown that our method works well up to 256 dimensions in M, X and Y each (equivalent to 768 total neurons).
>
> 2. The method departs from the ground truth at 512 dimensions; we will identify the reason and provide an update.
>
> 3. The computational complexity is O(ND^3) for the various matrix operations, where D=d_M+d_X+d_Y, and N is the number of iterations until convergence. We also provide a timing analysis in Fig 7 (attached PDF).
>
> 4. Our method does not require the use of PCA, however, we used PCA to obtain more stable estimates of the covariance matrix, and to minimize bias and error, as described above.
>
> 5. Our method also does not require particularly high firing rates: this is demonstrated in the multivariate Poisson simulation, where M1 and M2 had mean “spike-counts” of 2 (see Fig. 4a in the attached PDF). However, we expect that as firing rates increase, the data will appear more Gaussian, as a result of which our method will become more accurate. We will include a mention of this in the revised paper.
>
> 6. Inhomogeneous Poisson processes are those whose mean firing- (or “emission-”) rates change with time. Our method for estimating the PID does not consider the temporal characteristics of the original signal. Rather, the data analyst can choose the random variables M, X and Y to span some time range (or different time ranges) as they please. In our example, we counted the number of spikes in a 50-125 ms window after stimulus onset. Even if the underlying spiking process was an inhomogeneous Poisson process, this spike count would be Poisson distributed, with a mean given by the integral of the emission rate over the fixed window. In general, while analyzing data and computing PID values, one will have to be aware of severe inhomogeneities (e.g., if the distribution changes not just in the same way in each trial, but changes across trials), and account for them separately. For example, in our analysis, we excluded time periods when the mice were not “engaged” in the task, as defined by not actively consuming rewards at a rate of at least 2 rewards per minute. We will add a discussion of inhomogeneity, as well as how to choose M, X and Y, to the paper or to the supplementary material.

---

> > ### Author Response · Authors · 2023-08-18
> > **Follow up comment to Reviewer wULj**
> >
> > We request Reviewer wULj to let us know if our responses have addressed their concerns, and to kindly consider whether our paper is now worthy of a better score.
> >
> > We also request the reviewer to get back to us soon, since we author responses will be closed after August 21st, 1pm EDT.
> >
> > In addition to the overall rebuttal and the reviewer-specific rebuttal, we would also like to draw the reviewer's attention to the analysis posted in an official comment at the top of this page, which extends our simulation in Example 10 to 1024 dimensions in each of M, X and Y (equivalent to 3072 total neurons). We believe this addresses the reviewer's central point about extending our method to thousands of neurons, as explained in the official comment.

---

### Official Review · Reviewer_Tp5P · 2023-07-20

**Soundness:** 3 good
**Presentation:** 3 good
**Contribution:** 3 good
**Rating:** 6
**Confidence:** 4

**Summary:**

This article proposes an upper bound on mutual information (more precisely: on the "unique information" / "union information" that appear in "Partial Information Decomposition") that is easier to compute. This is done by replacing an infimum over all distributions matching given marginals by an infimum over only the ones that are Gaussian.
Toy examples (with Gaussian distributions) are given, and an experiment on real data is performed.
Further theory also includes considerations about biases arising from covariance estimators.

**Strengths:**

The paper is mostly clearly written, with simple examples to help the reader follow.
The paper is self-complete, with reminders of definitions.
Considering covariance estimator biases is a plus.

**Weaknesses:**

EDIT: after discussion with the authors, I see that I had misunderstood the scope and contributions of the paper. The weaknesses below do not longer hold (or not as strongly).

-----------

The main issue is that the definition of the upper bound actually supposes that the distribution MXY that is studied is Gaussian, at least marginally. Indeed the infimum is performed over all possible distributions Q_MXY that are Gaussian and whose marginals satisfy Q_MX = P_MX and Q_MY = P_MY, which implies that P_MX and P_MY have to be Gaussian (otherwise the infimum is performed over an empty set).
As a consequence, this upper bound definition cannot be applied to datasets that are not Gaussian (at least marginally). This crucial point is not discussed in the paper.

Actually, from Sections 3.2 and 5, it seems that the upper bound is estimated based on covariance matrices only, without using distribution Gaussianity. If this is the case indeed, then the definition could be changed, to rely directly on these covariance matrices and removing the Gaussianity assumption. This however would probably require to significantly rewrite the paper.


Another issue is the lack of sufficient validation, either theoretically or experimentally, in particular regarding distributions that are not Gaussian (i.e., really not Gaussian). This comment could have been tempered if there had been a discussion about general approximative Gaussianity of distributions in the field of study, but there is none.
Also, the impact of the introduction of such a Gaussianity constraint in the upper bound estimation should be studied closely: how realistic is this assumption, how far are real optimal Q_MXY from their Gaussian versions, how tight is the upper bound thus obtained, etc.

If this was a theoretical paper, I would expect theoretical results (guarantees such as a bound on the error made by the upper bound, or proving the conjectures, etc.). If this was an experimental paper, I would expect an extensive validation on many distributions for which the actual mutual informations are estimated (brute force or known solutions) in settings as varied as possible (not just normal distributions + a single real dataset). But the paper, in its current state, does meet these expectations.



**Questions:**

I would be willing to revise my score if the points in the section above were significantly addressed.

EDIT: they were.

**Limitations:**

Cf the validation issue in the weakness section above.

---

> ### Author Rebuttal · Authors · 2023-08-10
>
> **Weaknesses:**
>
> > The main issue is that the definition of the upper bound actually supposes that the distribution MXY that is studied is Gaussian, at least marginally … This crucial point is not discussed in the paper.
>
> It is true that our definition applies only to Gaussian P_MXY (in fact, it must be jointly Gaussian, not just marginally), and that the definition technically does not apply to non-Gaussian distributions. This is why we include the word “Gaussian” in the title of the paper.  We mention “we compute/estimate the PID on Gaussian distributions”, in the abstract (lines 9-11) and in the introduction (lines 48-49). We also make it clear that M, X and Y are jointly Gaussian just above Definition 2 (lines 112-113), and again in lines 132-133. Overall, we believe we have tried to convey the point that P_MXY is assumed to be jointly Gaussian.
>
> Nevertheless, we understand that there is a potential for confusion between the Gaussianity of P_MXY and the assumption restricting the optimization variable, Q_MXY, to be Gaussian. To make the Gaussianity of P_MXY more explicit, we will state this assumption within Definition 2 itself, rephrase line 120, and we will also explicitly make the point that the definition does not apply to non-Gaussian distributions, strictly speaking. We will also make small edits as necessary throughout the whole paper to ensure that this distinction is clear. We welcome any suggestions that may help make this point more clearly.
>
> Even though our method is designed only for Gaussian distributions, we believe it is an important contribution. Gaussian distributions have historically been a starting point for many estimators (e.g., correlation is used as a measure of dependence, but is exact only in the Gaussian case). Our ~-PID estimator is essentially the best in the field in terms of providing ground-truth validation at high dimensionalities; this was made possible because of Gaussianity. While the central claims and results of the study focus on Gaussian distributions, we showed that when distributions are close to Gaussian (e.g., Poisson) our method for computing PIDs does not immediately break down.
>
> > Actually, from Sections 3.2 and 5, it seems that the upper bound is estimated based on covariance matrices only … definition could be changed … would probably require to significantly rewrite the paper.
>
> As the reviewer correctly notes, our definition depends only on the covariance matrix, and can thus be applied to any data, even if the data itself is non-Gaussian. We do not believe this requires a rewrite of the paper; rather, we just apply an estimator that was designed for Gaussian distributions to data that is non-Gaussian. This is what we do in section 6: while Gaussian P_MXY forms the main scope of the paper in Sections 1-5 (please see the overall rebuttal), we justify the applicability of our method to spiking data using a Poisson simulation.
>
> > Another issue is the lack of sufficient validation, either theoretically or experimentally, in particular regarding distributions that are not Gaussian …
>
> Given that the main scope of the paper in Sections 1-5 is Gaussian distributions, we believe we have demonstrated sufficient empirical validation on Gaussian P_MXY. This has been shown through Examples 5-10 and Figures 1-4. Compared to other papers, we believe our paper provides one of the most comprehensive ground-truth validations (please see the overall rebuttal for a detailed comparison with [17] and [21]).
>
> However, we agree that it is important to justify that our method can be applied to non-Gaussian neuroscientific data, which we do in Section 6 and in the additional analyses (please see overall rebuttal). However, we do not make claims in general non-Gaussian cases, which could be very far from Gaussian. We will make this explicit in the revised paper. While a broader and more rigorous analysis of non-Gaussian distributions is important, we believe it is beyond the scope of the current paper.
>
> > … If this was an experimental paper, I would expect an extensive validation on many distributions for which the actual mutual informations are estimated (brute force or known solutions) …
>
> This is an empirically-focused paper, and we do have extensive validation for Gaussian distributions, which is the main scope of the paper.
>
> As the reviewer states, comparisons on a wider variety of Gaussian and non-Gaussian examples are limited by the availability of ground truth. This is why we rely on the MMI-PID (Definition 3) and Barrett’s theorem [13] (lines 177-178), which provides a closed form expression for the ~-PID for Gaussian P_MXY, _for scalar M_. Using this, we show that we can recover the ground truth by restricting Q_MXY to be Gaussian in Examples 5-7 (Fig 1). We then construct more complex examples using additivity (Property 1), and show that even then, the restriction to Gaussian Q_MXY can recover the ground truth (Examples 8-10; Figs 2, 3). To go beyond these examples, we do not know of brute-force methods for computing the ~-PID for Gaussian P_MXY, except for [17], which applies to general distributions P_MXY with scalar M, X and Y (which is trivial for Gaussian P_MXY due to Barrett’s result), and [21], which is not sufficiently well-tested to consider as “ground truth”, and was also published too recently for us to perform comparisons. For the same reasons, comparisons with non-Gaussian distributions at higher dimensions (beyond small Poisson/Binomial) are also limited by the availability of ground truth.
>
> > I would be willing to revise my score...
>
> To summarize:
> - We will make the distinction between the Gaussianity of P and Q clear
> - For the scope of Gaussians, we believe our evaluations are comprehensive
> - We have performed additional analyses on non-Gaussian distributions, and we will be explicit that our method does not extend in general
>
> Please let us know if our arguments and updated results are convincing, and how we may improve our score.

---

> > ### Comment · Reviewer_Tp5P · 2023-08-11
> >
> > Thank you for your reply. I had misunderstood the scope of the paper indeed. I thought the Gaussian case was trivial and that the point was to reduce non-Gaussian examples to Gaussian ones as a first approximation (through Definition 2). Now I understand that even the Gaussian case is complex.
> >
> > I thought that for Gaussian distributions, the minimizer Q in Definition 1 would necessarily be Gaussian, and thus introducing Definition 2 was relevant only for non-Gaussian distributions (hence the misunderstanding on the scope).
> >
> > If the conjecture is true, i.e. that minimizers Q in the Gaussian case are Gaussian indeed, then the two definitions are identical. If I understand well, the contribution of the paper in that case is not to bring a new definition, but to make explicit use of the knowledge that the minimizer is Gaussian, to help the optimization process, while this was not done in previous papers. Am I right?

---

> > > ### Author Response · Authors · 2023-08-13
> > > **Response to Comment by Reviewer Tp5P**
> > >
> > > We are very grateful to the reviewer for their response.
> > >
> > > > I had misunderstood the scope of the paper indeed. I thought the Gaussian case was trivial and that the point was to reduce non-Gaussian examples to Gaussian ones as a first approximation (through Definition 2). Now I understand that even the Gaussian case is complex.
> > > >
> > > > I thought that for Gaussian distributions, the minimizer Q in Definition 1 would necessarily be Gaussian, and thus introducing Definition 2 was relevant only for non-Gaussian distributions (hence the misunderstanding on the scope).
> > >
> > > The reviewer is absolutely correct that the Gaussian case is also non-trivial in general. They also correctly recognize that it is _not known_ whether the minimizer Q is guaranteed to be Gaussian if P_MXY is jointly Gaussian. Even for Gaussian P_MXY, this is only known in a few specific cases such as when M is scalar, or if P_MXY satisfies a condition shown in [12].
> > >
> > > > If the conjecture is true, i.e. that minimizers Q in the Gaussian case are Gaussian indeed, then the two definitions are identical. If I understand well, the contribution of the paper in that case is not to bring a new definition, but to make explicit use of the knowledge that the minimizer is Gaussian, to help the optimization process, …
> > >
> > > Indeed, we do _not_ provide a new definition. Rather, the central contribution of our paper is to develop and evaluate an efficient method to estimate the PID for Gaussian P_MXY, by restricting the space of the minimizer Q to Gaussian variables. As the reviewer states, this would result in the correct solution if our conjecture is true. By assuming Q to be Gaussian, we can make use of the properties of Gaussian distributions (e.g., closed-form expressions for mutual information) to develop an efficient projected gradient descent optimizer, and then check how well we perform (please see _”Evaluating the Gaussianity assumption”_ below).
> > >
> > > > … while this was not done in previous papers. Am I right?
> > >
> > > A previous paper by Venkatesh and Schamberg [12] used a similar technique of restricting Q to be Gaussian for computing a different PID definition called the delta-PID. Our work makes significant advances over [12], on multiple fronts:
> > >
> > > 1. We compute a different PID definition, the ~-PID, which satisfies better properties than the delta-PID used in [12]. Most importantly, the ~-PID satisfies a very fundamental property called **additivity** (please see our response to reviewer DDWz on why additivity is fundamental).
> > >
> > > 2. We examine several examples (Examples 5-10), increasing in complexity and dimensionality, to show that restricting Q to be Gaussian is reasonable, and allows us to recover the ground truth. [12] only tested this for the simplest case of Gaussians with scalar M, and fails in more complex cases (Example 8; Fig 2 in the paper).
> > >
> > > 3. Our method is faster (please see Fig 7 in the PDF attached to the overall rebuttal).
> > >
> > > 4. Our method is capable of computing higher dimensionalities (as described in our response to reviewer DDWz).
> > >
> > > 5. Finally, we consider the problem of _estimation_, i.e., computing the PID from a covariance matrix that is estimated from real data, and show how bias in PID estimates can be corrected. This is not addressed in [12].
> > >
> > > These points are summarized in the overall rebuttal near the top of this page.
> > >
> > > **Evaluating the Gaussianity assumption**
> > >
> > > Since we do not have a guarantee that the minimizer Q is actually Gaussian for all Gaussian P_MXY, we test what the reviewer asked in their original review, but for _Gaussian_ P_MXY. That is, we test: “how realistic is this assumption [of restricting Q to be Gaussian], how far are real optimal Q_MXY from their Gaussian versions”. We consider a number of examples of increasing complexity (Examples 5-10), where the ground truth PID values are known (starting with scalar M [13], and then using the additivity property). We show that our method is able to recover the ground truth, proving that for all the examples we consider, the optimal Q _is_ in fact Gaussian, and hence our conjecture still stands.
> > >
> > > Incidentally, our method is also applicable to non-Gaussian P_MXY, because it only relies on the covariance matrix, as noted by the reviewer in their original review. So we further tested whether it was reasonable to apply our method to Poisson distributions (now extended to more non-Gaussian distributions; see overall rebuttal). Here, we do not have ground truth, so we instead compared our results with another ~-PID estimator [20] (which works only for discrete distributions with limited support). We showed that our PID estimator with Q-restricted-to-be-Gaussian comes very close to the discrete estimator of [20] on a multivariate Poisson distribution P_MXY (Fig 5 in the paper). We then used this as a basis to apply our PID estimator to real neural data, and demonstrated its utility in providing new insights about interactions between brain regions (Fig 6 in the paper).

---

> > > > ### Comment · Reviewer_Tp5P · 2023-08-16
> > > >
> > > > Thank you for your detailed answer.
> > > >
> > > > Since complexity and dimensionality are relevant in the claims, I have questions regarding the optimization.
> > > >
> > > > 1. From the last figures you have provided, it looks like the computational complexity is cubic. In the paper, it seems that 2 matrix inverses (or pseudo-inverses) and one SVD need to be perform, which would explain this cubic complexity. Do you confirm? Where are the computational bottlenecks? It could be relevant to add a line in the paper stating that complexity.
> > > >
> > > > 2. How stable is the computation of the gradient? In particular, the projection (Equation 13 + Appendix lines ~470-500) replaces negative eigenvalues by 0. What about eigenvalues that are close to 0? Are there very small eigenvalues that could be on either side or the 0 threshold, up to statistical fluctuations (such as +-1 sample), and what is the impact of such variations (as they change the dimension of the projection subspace, and consequently the projected matrix as well)? Actually, have you checked the spectrum?

---

> > > > > ### Author Response · Authors · 2023-08-17
> > > > > **Response to Comment #2 by Reviewer Tp5P**
> > > > >
> > > > > > From the last figures you have provided, it looks like the computational complexity is cubic. In the paper, it seems that 2 matrix inverses (or pseudo-inverses) and one SVD need to be perform, which would explain this cubic complexity. Do you confirm? Where are the computational bottlenecks? It could be relevant to add a line in the paper stating that complexity.
> > > > >
> > > > > Yes, the computational complexity of each gradient descent iteration is cubic in the dimensionality of M, X and Y. Suppose for simplicity that M, X and Y all have the same dimensionality $d$. Then, the computational complexity of the objective, the gradient and the projection operator are all $\mathcal O(d^3)$. Since each of these steps has a similar complexity, we don’t believe that there is any particular bottleneck.
> > > > >
> > > > > We will certainly add a statement about computational complexity to the revised paper, and include the timing analysis from the attached PDF as well.
> > > > >
> > > > > > How stable is the computation of the gradient? In particular, the projection (Equation 13 + Appendix lines ~470-500) replaces negative eigenvalues by 0. What about eigenvalues that are close to 0? Are there very small eigenvalues that could be on either side or the 0 threshold, up to statistical fluctuations (such as +-1 sample), and what is the impact of such variations (as they change the dimension of the projection subspace, and consequently the projected matrix as well)? Actually, have you checked the spectrum?
> > > > >
> > > > > **Regularization for stable inverses:** The main source of instability in computing the gradient are the matrix inverses in Eq (55) in the Appendix. These are regularized by adding an identity matrix multiplied with 1e-7 (as mentioned in the Appendix, lines 491-492).
> > > > >
> > > > > We are not completely sure what the reviewer means regarding the eigenvalues, but we try to offer a response based on our best understanding: we first note that negative eigenvalues are set to zero on $\Sigma_{XY|M}$, whereas the optimization variable $\Sigma$ is an off-diagonal block of this matrix (please see Eq (56)). Setting several eigenvalues to zero in $\Sigma_{XY|M}$ will also affect the rank and the invertibility of the matrix $S = I - \Sigma\Sigma^T$, whose inverse appears in Eq (55). This is why we have the regularization in place: $S^{-1} = ((1 + \epsilon)I - \Sigma\Sigma^T)^{-1}$, with $\epsilon = 10^{-7}$.
> > > > >
> > > > > In the projection step, we only set negative eigenvalues to zero. We do not apply a threshold to zero-out any small positive eigenvalues. We have checked the spectrum of $\Sigma_{XY|M}$, and it is certainly possible for it to have several small positive eigenvalues. Along with the eigenvalues set to zero, this may affect the rank and invertibility of $S$, as mentioned above. However, we observed that the aforementioned regularization solves this problem in practice. We have now also added an extra regularization term to the projection operator for the matrix inverses in Eq (58).
> > > > >
> > > > > **Role of RProp:** We also note that RProp is designed to suppress the learning rate of any components along which the gradient is unstable and/or fluctuating a lot. Since fluctuations will likely change the sign of the respective gradient component rapidly, RProp will automatically suppress the learning rate $\eta$ along that component specifically (please see Eq (61)).
> > > > >
> > > > > **Not using SGD:** Please also note that this is _not_ a stochastic gradient descent method. The objective is fixed for a given covariance matrix (even if the covariance itself is estimated from data), and there is no concept of a mini-batch, so the stability of the gradient under stochasticity of mini-batches is not a concern (in case this was a point of confusion).
> > > > >
> > > > > **Ground truth comparison is the ultimate test:** Ultimately, the true test of whether our implementation works lies in whether it recovers ground truth. We have shown that our method recovers ground truth in various cases (Examples 5-10), including when M, X and Y each have a dimensionality of 1024 (please see the updated result in an official comment at the top of this page).
> > > > >
> > > > > We request the reviewer to let us know if we have answered their questions, and if they would kindly consider raising their score, based on a renewed understanding of the scope and contributions of our paper.

---

> > > > > > ### Comment · Reviewer_Tp5P · 2023-08-17
> > > > > >
> > > > > > > We request the reviewer to let us know if we have answered their questions, and if they would kindly consider raising their score, based on a renewed understanding of the scope and contributions of our paper.
> > > > > >
> > > > > > Yes, thank you for all your answers, I will raise my score.
> > > > > > A last question, regarding RProp: the paper (and the comment above) mention RProp (which dates from 1992), but the paper referred to ([33]) is RMSProp (2012). Which one do you use? To me, they are very different techniques.

---

> > > > > > > ### Author Response · Authors · 2023-08-17
> > > > > > > **Response to Comment #3 by Reviewer Tp5P**
> > > > > > >
> > > > > > > We are extremely grateful to the reviewer for their time and effort in providing a helpful review, and for having a constructive discussion with us. We also thank the reviewer for raising their score.
> > > > > > >
> > > > > > > > A last question, regarding RProp: the paper (and the comment above) mention RProp (which dates from 1992), but the paper referred to ([33]) is RMSProp (2012). Which one do you use? To me, they are very different techniques.
> > > > > > >
> > > > > > > We thank the reviewer for raising this point. We do use RProp, not RMSProp, and we agree that they are different. We will use the correct reference to _Riedmiller and Braun (1993)_ shown below, to make this clear.
> > > > > > >
> > > > > > > Riedmiller, Martin, and Heinrich Braun. "A direct adaptive method for faster backpropagation learning: The RPROP algorithm." _IEEE international conference on neural networks_. IEEE, 1993.
> > > > > > >
> > > > > > > We had been unable to find this reference at the time of submission, so we had cited [33] instead, which points to a set of lecture notes and a webpage that include an explanation of RProp.

---

### Official Review · Reviewer_jMyH · 2023-07-25

**Soundness:** 3 good
**Presentation:** 4 excellent
**Contribution:** 2 fair
**Rating:** 7
**Confidence:** 3

**Summary:**

First, I would like to thank the authors for the work they put into contributing to the field.
In their paper, the authors provide a new method to estimating a well-known measure, the Partial Information Decomposition (PID). Within the topic of PIDs, their method computes a specific version of PID, the ~-PID.

While methods to compute the ~-PID already exist, the authors' method is advantageous in that
1) it is more efficient to compute (quadratic in number of variables instead of exponential) and
2) in that it corrects for the bias.

The author's then go on to show with simulations and examples that
1) their choice of PID method, the ~-PID, outperforms other PID methods
2) the bias correction works empirically
3) their method is stable even for increasing dimensionality
4) their method can be applied on data which is not Gaussian distributed such as simulated Poisson data and real neural recordings (neuropixel)

**Strengths:**

- The paper addresses a topic which is becoming more and more relevant as the number of neurons that can be recorded from simultaneously is increasing: The question of how information is distributed among neurons and brain areas.
- The paper is written in a very clear and structured way which successfully guides the reader through every logical step.
- The mathematical formulation is sound
- The quality of figures is excellent

**Weaknesses:**

- The paper is an improvement of an already existing method which makes it important and publishable yet disqualifies it for a very high grade.
- The claim of generalizability to non-Gaussian data is not backed up strongly enough empirically or mathematically (refer to "Questions")
- Small mistakes/typos in lines: 177, 206, 322

**Questions:**

- In the list of contributions, the authors state in line 50 that they are able to reduce the number of optimization variables from exponential to quadratic. Where can I see this in the equations and derivations?

- The authors show that their method works on non-Gaussian data by applying it to simulated Poisson data and neuropixel spike count data. They explain this by stating that the Gaussian distributions is a good-enough approximation to the Poisson distribution. It is known, however, that neural firing does not follow a Poisson distribution. This is especially pronounced when recorded via 2photon imaging for which the distribution is strongly inflated at zero. Xue-Xin Wei et al. 2020 propose a zero-inflated gamma model to accurately capture calcium imaging traces. Is it feasible for the authors to test their method on such bimodal distributions which might be harder to approximate with a Gaussian distribution? If the rebuttal time for such experiments is too short, can the authors comment on the expected outcome?

**Limitations:**

The authors discuss and point out most limitations. The claim that their method works on non-Gaussian data is only backed by an experiment with Poisson data, however. In my opinion, there is still room for the method to fail for other relevant distributions (refer to point "Questions") which would be worth mentioning as a (possible) limitation.

---

> ### Author Rebuttal · Authors · 2023-08-10
>
> **Weaknesses:**
>
> > The paper is an improvement of an already existing method which makes it important and publishable yet disqualifies it for a very high grade.
>
> We thank the reviewer for raising this concern, and providing us with an opportunity to explain why we believe our advance is significant:
>
> 1. In the overall rebuttal, we explain how our advance is significant with respect to other works in the field, and in particular the work of Venkatesh et al. (2022). In particular, our method works at higher dimensions, runs much faster and corrects for bias.
>
> 2. Additivity is an extremely fundamental property: effectively, it states that the PID values of an isolated system should not depend on the PID values of another isolated system (please see our response to reviewer DDWz for an explanation). The ~-PID satisfies the additivity, whereas the delta-PID does not. This makes our method much more practically applicable than the method based on the delta-PID. Apart from additivity, several works have shown that the ~-PID also has other strong operational motivations (e.g., Kolchinsky, Entropy 2022), which has not been shown to the same degree for the delta-PID. These points (additivity in particular), make it extremely important to have a method capable of computing the ~-PID. We will add an explanation of the importance of additivity to the revised version of our paper.
>
> 3. Bias correction is extremely important for obtaining correct PID values, as evidenced by the sheer amount of bias seen at small sample sizes (Fig. 4, and Figs. 9 and 10 in the appendix). For example, we find that synergy is often highly over-estimated. To our knowledge, ours is the first paper to demonstrate the severity of not correcting for bias when estimating PIDs from real data, and we are also the first to propose and evaluate a method to correct this bias. We will re-emphasize the importance of bias correction in the revised paper.
>
> > The claim of generalizability to non-Gaussian data is not backed up strongly enough empirically or mathematically (refer to "Questions")
>
> Thank you for raising this point. We did not mean to claim generalizability to non-Gaussian distributions in general. Rather, since our Gaussian PID method uses only a covariance matrix, we tried to justify its application to non-Gaussian spiking neural data before demonstrating its utility in a practical neuroscientific setting. Accordingly, we presented a small multivariate Poisson simulation (which was the largest possible example in which we could compare with known estimators), which we have now extended to include a few more non-Gaussian cases. We hope our new simulations highlight the limitations of the applicability of our method to non-Gaussian data, and show that care needs to be exercised when drawing interpretations from the results of our method. A more detailed response is given below.
>
> > Small typos ...
>
> Thank you, we will correct these.
>
> **Questions:**
>
> > In the list of contributions, the authors state in line 50 that they are able to reduce the number of optimization variables from exponential to quadratic. Where can I see this in the equations and derivations?
>
> We thank the reviewer for raising this question, and we will add the following explanation to the paper:
> The general form of the ~-PID involves optimizing over Q_MXY. If we consider M, X and Y to be discrete, with a support of size K in each dimension, then the number of degrees of freedom of Q will be O(K^(d_M + d_X + d_Y)). For general continuous distributions, if we discretize the distribution, the same analysis would hold. In the case of our Gaussian ~-PID method, the optimization variable is \Sigma_{X,Y|M}, which has a dimensionality of just d_X * d_Y.
>
> > The authors ... applying it to simulated Poisson data ... It is known, however, that neural firing does not follow a Poisson distribution. This is especially pronounced when recorded via 2photon imaging for which the distribution is strongly inflated at zero. ... Is it feasible for the authors to test their method on such bimodal distributions ...?
>
> We agree that further validation on non-Gaussian data to understand the impact of non-Gaussianity is important. As described in the overall rebuttal, we have now included an example on a multivariate Binomial distribution (which is also close to Gaussian), as well as zero-inflated versions of both the multivariate Poisson and multivariate Binomial distributions. Our observations are outlined in the overall rebuttal.
>
> We note that a more extensive evaluation of non-Gaussian distributions (especially at higher dimensionalities) is made difficult by the unavailability of ground truth. The “ground truth” in our multivariate Poisson example is obtained using the discrete ~-PID estimator of Banerjee et al. (which we assume is more accurate for discrete variables). Our method provides a better result than the delta-PID and the MMI-PID, at least part of which could be attributed to the difference in definition. Nevertheless, we believe it demonstrates that applying our method to non-Gaussian data need not be ruled out, although care should always be taken in interpreting results.
>
> We will add both the new simulations, as well as this discussion on careful interpretation to the revised version of our paper.
>
> With the addition of new results showing the extent to which our method applies to non-Gaussian data, we request the reviewer to reconsider whether our paper meets the criteria for a higher score.

---

> > ### Comment · Reviewer_jMyH · 2023-08-14
> > **Clarifications and extra experiments**
> >
> > I thank the authors for their clarifications concerning the impact of the work and for their additional experiments showing the (limited) generalizability to non-Gaussian data. I have raised the score from 6 to 7.

---

### Author Rebuttal · Authors · 2023-08-10

We sincerely thank all of the reviewers for taking the time and effort to provide thoughtful and constructive reviews of our work.

We use this space to address a few comments that were common across reviewers, and to explain the additional analyses we perform in response to these comments. We also reiterate the scope and the central contributions of our paper, and describe how these are laid out in the paper.

**Summary of new analyses (and figure numbers in the attached PDF)**

1. Increased dimensionality in Example 10 (Fig 5)
2. Increased number of PCA components in neural data analysis (Fig 8,9). Please see response to reviewer wULj for details.
3. Comparison with delta-PID at higher dimensions (Fig 6,7)
4. More non-Gaussian examples: Binomial, and zero-inflated Binomial and Poisson (Figs. 1-4)

**Scope of the paper**

Our paper provides a new method to compute and estimate the ~-PID definition for multivariate, jointly Gaussian distributions P_MXY. This is done by assuming Gaussian optimality in the optimization problem used to compute the PID, which helps reduce the number of optimization variables. To justify the Gaussian optimality assumption, we consider a number of Gaussian examples with known ground truth, and show that the output of our method agrees with ground truth, even at high dimensionalities. The aforementioned are the subject of Sections 1-5, which deal strictly with Gaussian P_MXY, and endeavor to demonstrate the efficacy of our method on Gaussian distributions. However, since most real data is non-Gaussian, in Section 6, we showed using a low-dimensional simulation, that our method also gives reasonable results on non-Gaussian (Poisson) data. We then apply it to a publicly available neuroscientific dataset to investigate the amount of redundancy between different visual cortical brain regions.

**Extending our method to higher dimensionalities**

In the submitted paper, we presented evidence of agreement with ground truth only up to the dimensions of M, X and Y each being 128 (i.e., a total of 384 dimensions for a covariance matrix of size 384x384). We have now extended this analysis to higher dimensions (Fig. 5 in the attached PDF), showing that our method continues to agree with ground truth up to M, X and Y each being 256-dimensional, i.e., a total of 768 dimensions, for a covariance matrix of size 768x768 (see new figures in attached PDF). Our method begins to deviate from the ground truth at 512 (at which point the covariance matrix has 2.36 million elements). We will investigate the root issue of this problem and provide an update within a week.

It should be noted that our method is the first to provide PID estimates that agree with ground truth at such high dimensions, and that most other works do not come close (see comparisons below).

**More non-Gaussian examples**

In the submitted paper, we only tested our method on a single non-Gaussian, multivariate Poisson example. We have now extended this to include a multivariate Binomial example, as well as zero-inflated versions of the multivariate Poisson and Binomial setups (Fig 1-4 in the attached PDF). We find that our method performs well on the original Poisson and Binomial versions, agreeing with the discrete PID estimator of Banerjee et al. (ISIT 2018).

In the zero-inflated versions, when we have a bimodal distribution with a point-mass at zero, our Gaussian PID estimator no longer recovers the same _absolute_ PID values as the estimator by Banerjee et al.. However, when normalized by the total mutual information, the _relative_ PID-values are still close to the values computed by the Banerjee et al. method and follow the correct trends in every case. Thus, for such a distribution, an improved estimate of the mutual information (e.g., Kraskov et al. 2004; Belghazi et al. ICML 2018) could potentially be used to correct the absolute PID values, after a more thorough evaluation.

The primary challenge with testing non-Gaussian distributions is the absence of good ground truth. We currently rely on a method from Banerjee et al. (2018), designed for discrete distributions, as ground truth. We cannot be sure if the errors seen in the ultimate plot are a result of errors in our method, or errors in computing the ground truth itself.

**Improvements over the delta-PID & other PID measures**

Many reviewers asked us to expand upon how our work improves on prior art. We provide a summary below, which we will include in the Related Work section of our paper. Also, we have now compared our method with the delta-PID at higher dimensions (Fig 6,7 in the attached PDF).

We divide previous works into 3 categories:
1. Compared to Venkatesh et al. (ISIT 2022), here we have:
- Better ground truth validation
- Ability to compute at higher dimensionality (256, rather than 64 or 128)
- Faster computation (>1000x faster at d_M=d_X=d_Y=64)
- Satisfy additivity (which is important; see response to reviewer DDWz)
- Correct for bias in estimates
2. Compared to discrete PID estimators (Banerjee et al., ISIT 2018; Makkeh et al., Entropy 2018), we have
- Much higher dimensionality: both discrete PID estimators were demonstrated for discrete M, X and Y with support sizes of at most 18 in Makkeh et al., which corresponds to a total support of 18^3=5832 for Q_MXY. If we assume a single neuron can have at most 10 spikes (i.e., support of 10), then these methods can handle ~4 neurons, since log_10(5832) < 4.
- We discuss estimation and how bias can be corrected
3. Compared to the continuous PID estimators of Pakman et al. [17] and Liang et al. [21], we have:
- An estimator that matches ground truth at high dimensionalities. [17] shows ground truth validation only in scalar cases, while [21] matches ground truth only for scalar binary cases and deviates from ground truth in other examples.
- We demonstrate our method on higher dimensionalities compared to [17].
- We show bias correction

---

### Author Response · Authors · 2023-08-15
**Extending our method to a dimensionality of 1024**

Dear reviewers,

We were able to extend the dimensionality of our method to 1024 in Example 10 (results shown below). The dimensionality here refers to the number of dimensions in each of the vectors, M, X and Y. This is therefore equivalent to having a total of 3072 neurons across three brain regions, or 1024 PCA components in each brain region, with a covariance matrix of size 3072x3072.

We believe this addresses Reviewer wULj's question of whether our method can achieve the stated goal of being able to analyze thousands of simultaneously recorded neurons. Although this is a simulated example and not real neural data, we believe this example is more informative about our method's capabilities since it has known ground truth and allows for computing error metrics (as shown below).

This also expands the difference between the capabilities of our method and the delta-PID of [12] (as requested by Reviewer DDWz), which failed at a dimensionality of 256 on the same example, and ran much slower (our method takes roughly the same amount of time at 1024 dimensions, as theirs takes at 64 dimensions).

The earlier error in our method at 512 (in the PDF attached to the overall rebuttal) stemmed from the projection step of our projected gradient descent, where we had missed adding a regularization term while inverting a matrix. This did not cause problems until a very high dimensionality, since the condition number of the matrix being inverted got worse with dimensionality. We have now added an adaptive regularization, which ensures that the projection step works correctly at any dimensionality.

**Updated Results**

The following table shows the results from Example 10 (Fig 3 in the paper), extended to 1024 dimensions in each of M, X and Y. The results are shown for successively doubling at a gain value of 3.0 in X_1 (i.e., for successively doubling the last data point in Fig 2(left) in the paper). The errors shown are taken with respect to ground truth, and represent the maximum error across all PID components.

We have not yet tried running this analysis at a higher dimensionality of 2048, since we wanted to share this update as soon as possible.

| Dimension | I(M;(X,Y)) | UI_X | UI_Y | RI | SI | Max absolute error (bits) | Max relative error | Compute time (s) |
|-----------|-------------|------|------|----|----|----------------------|----------------|------------------|
| 2 | 2.73 | 1.16 | 0.29 | 1.0  | 0.28 | 6.49e-08 | 2.35e-07 | 0.04 |
| 4 | 5.46 | 2.32 | 0.58 | 2.0  | 0.55 | 1.26e-07 | 2.28e-07 | 0.04 |
| 8 | 10.92 | 4.64 | 1.17 | 4.0  | 1.11 | 2.41e-07 | 2.18e-07 | 0.05 |
| 16 | 21.84 | 9.29 | 2.34 | 8.0  | 2.21 | 4.97e-07 | 2.25e-07 | 0.06 |
| 32 | 43.68 | 18.58 | 4.68 | 16.0 | 4.42 | 1.01e-06 | 2.29e-07 | 0.11 |
| 64 | 87.35 | 37.15 | 9.36 | 32.0 | 8.84 | 2.05e-06 | 2.32e-07 | 0.29 |
| 128 | 174.70 | 74.30 | 18.72 | 64.0 | 17.68 | 4.13e-06 | 2.34e-07 | 1.11 |
| 256 | 349.40 | 148.60 | 37.44 | 128.0 | 35.36 | 8.28e-06 | 2.34e-07 | 3.59 |
| 512 | 698.81 | 297.21 | 74.88 | 256.0 | 70.73 | 1.66e-05 | 2.35e-07 | 59.14 |
| 1024 | 1397.61 | 594.41 | 149.75 | 512.0 | 141.45 | 3.32e-05 | 2.35e-07 | 473.42 |

We request the reviewers to consider whether these results (in addition to the extra analyses in the overall rebuttal) make our paper worthy of a higher score.

---

### Decision · Program_Chairs · 2023-09-21

**Decision:**

Accept (spotlight)

**Comment:**

This paper makes significant contributions to an important topic: that of estimating partial information decompositions, which is a complex task in high dimension.
Compared to previous works, the author's method scales better with dimension and is bias-corrected.
The theory presented in this paper is sound.
All reviewers noted the clarity of the writing of the paper and that of the experiments.

This paper has a positive consensus, as all five reviewers argue for acceptance of the paper.

One reviewer initially argued for the rejection of the paper, yet the authors managed to convince them of the merits of their approach with a very detailed rebuttal. The added experiments run during the discussion period will enhance an already great paper.

In short, the consensus is that this paper will have a practical impact on brain signal processing.